# RNA helicase DDX3X modulates herpes simplex virus 1 nuclear egress

Bita Khadivjam[1], Éric Bonneil [2], Pierre Thibault[2,3] & Roger Lippé [1,4✉]

DDX3X is a mammalian RNA helicase that regulates RNA metabolism, cancers, innate immunity and several RNA viruses. We discovered that herpes simplex virus 1, a nuclear DNA replicating virus, redirects DDX3X to the nuclear envelope where it surprisingly modulates the exit of newly assembled viral particles. DDX3X depletion also leads to an accumulation of virions in intranuclear herniations. Mechanistically, we show that DDX3X physically and functionally interacts with the virally encoded nuclear egress complex at the inner nuclear membrane. DDX3X also binds to and stimulates the incorporation in mature particles of pUs3, a herpes kinase that promotes viral nuclear release across the outer nuclear membrane. Overall, the data highlights two unexpected roles for an RNA helicase during the passage of herpes simplex viral particles through the nuclear envelope. This reveals a highly complex interaction between DDX3X and viruses and provides new opportunities to target viral propagation.

[1] Centre de recherche du CHU Sainte-Justine, Montreal, Quebec, Canada. [2] Institute for Research in Immunology and Cancer, University of Montreal, Montreal, Quebec, Canada. [3] Department of Chemistry, University of Montreal, Montreal, Quebec, Canada. [4] Department of Pathology and Cell biology, University of Montreal, Montreal, Quebec, Canada. ✉email: roger.lippe@umontreal.ca

DDX3X is an ATP-dependent RNA helicase that belongs to the DEAD-box family[1]. It interacts with dozens of cellular proteins, regulates RNA metabolism, cell cycling, apoptosis and exhibits both oncogenic and tumour suppressor activities[2,3]. DDX3X is also an important player in the induction of innate immunity and cytokine/chemokine production[4–8]. Not surprisingly, DDX3X modulates several RNA viruses, with both pro- and anti-viral consequences[9–17]. However, the implication of DDX3X is not limited to RNA viruses, since it is incorporated in hepatitis B virions, negatively modulates the reverse transcription of its DNA genome and inhibits vaccinia virus spread[18,19]. DDX3X is also incorporated in at least three distinct herpesviruses, namely herpes simplex virus 1 (HSV-1), pseudorabies virus (PRV) and human cytomegalovirus (HCMV)[20–24]. Though we previously reported that DDX3X is needed for the optimal production of infectious HSV-1 particles by modulating, not surprisingly, viral gene transcription and translation[25], how or why the protein is incorporated in mature virions is unknown. DDX3X is thus a multifunctional protein that impacts both RNA and DNA viruses, which makes it an attractive therapeutic target.

HSV-1 is a member of herpesviruses that are associated with a panoply of human conditions[26]. It is an enveloped DNA virus that replicates its genome and assembles new capsids in the nucleus[27,28]. At least four types of viral particles are produced in the nucleus of infected cells. This starts with procapsids, which are filled with scaffold proteins, are thermo-unstable, free of nucleic acid and round. Upon cleavage, the scaffold is partially released from the capsid to make room for the viral genome while the capsid acquires an icosahedral shape, producing so-called C-capsids. These concomitant processes are not absolute and abortive viral intermediates are also produced, including isocahedral A-capsids devoid of the scaffold and B-capsids that still retain the scaffold[28]. C-capsids are preferentially exported out of the nucleus over A- or B-nuclear capsids but the mechanism driving this selectivity is not fully understood[29–36]. Since these C-capsids (125 nm) are too large to pass through nuclear pores, they reach the cytoplasm by an unconventional route[30] that first involves their budding through the inner nuclear membrane, which produces enveloped perinuclear virions. These short-lived virions then fuse with the outer nuclear membrane, a de-envelopment step that releases non-enveloped capsids into the cytoplasm. The capsids then acquire their final envelope from a cellular compartment before reaching the cell surface[37,38].

The $pU_L31$ and $pU_L34$ viral proteins are critical viral components that form a so-called nuclear egress complex (NEC) that promotes capsid nuclear budding through the inner nuclear membrane and stimulates vesicle formation in vitro[35,39–53]. NEC components are detectable on nuclear capsids, the nuclear envelope and perinuclear virions but are absent from mature extracellular viruses[21,36,43,54–56]. $pU_L31$ and $pU_L34$ are both substrates of the viral $pU_S3$ viral kinase whose enzymatic activity alters the smooth distribution of NEC components around the nucleus[43,45]. While these phosphorylation events were initially thought to prime the NEC for the subsequent release of the enveloped virions out of the perinuclear space[42,45,57–59], recent findings suggest that NEC phosphorylation may serve as a negative regulator of uncontrolled membrane budding until C-capsids are docked to the NEC[60–62]. Interestingly, deletion of $pU_S3$ does not prevent the initial budding step but rather causes the accumulation of enveloped perinuclear virions in nuclear herniations, pointing to the dual role of the viral kinase in the nucleus and perinuclear space during viral egress[42,45,63]. The cellular p32, TorsinA, SUN2, ESCRT III components ALIX and CHMP4B, VAPB, CD98, protein kinases C and D as well as SCL35E1 have additionally been implicated in herpesviruses

nuclear egress[64–74]. However many molecular details of this complex nuclear egress pathway remain to be clarified.

Given the incorporation of DDX3X in HSV-1 virions and its ability to modulate viral gene expression[21,23,25], we initially sought by fluorescent microscopy to define where the virus might encounter DDX3X. Though DDX3X predominantly accumulated in the cytosol of uninfected cells as expected, the virus surprisingly redirected the cellular protein to the nuclear envelope. RNA interference and EM studies hinted that this was functionally relevant since DDX3X depletion led to an accumulation of enveloped perinuclear virions in nuclear herniations similar to those reported for HSV-1 or PRV mutants lacking the viral $pU_S3$ kinase or those that simulate constitutively phosphorylated $pU_L31$[59]. To decipher how DDX3X might be involved in HSV-1 nuclear egress, we identified DDX3X binding partners by mass spectrometry from HSV-1 infected cells, which revealed 15 different viral proteins. Among them, $pU_L31$ was of particular interest given its role in nuclear egress and the nuclear relocalisation of DDX3X. Binding of DDX3X to $pU_L31$ was confirmed by co-immunoprecipitation, immunofluorescence and with viral mutants. Most intriguingly, DDX3X nuclear envelope localisation was not only NEC and $pU_S3$ dependent but also C-capsid dependent, while NEC nuclear localisation was independent of DDX3X. Moreover, DDX3X co-localised with large capsid foci at the nuclear periphery, whose presence was both DDX3X and C-capsid dependent, hinting at a functional link between DDX3X, C-capsids and the NEC. Further analyses by EM indicated those were perinuclear virions located between the two nuclear membranes. They also revealed that DDX3X was required for mature virions to incorporate normal levels of $pU_S3$ and that the two proteins interact on enriched nuclear membranes, explaining why DDX3X or $pU_S3$ depletion lead to similar phenotypes. Altogether, this indicates that DDX3X cooperates with the viral nuclear egress machinery and the $pU_S3$ viral kinase during HSV-1 while the virus exits the nucleus. These exciting findings highlight unexpected functions for a mammalian RNA helicase during the nuclear maturation of a DNA virus and will open up new research avenues.

## Results

**HSV-1 redirects DDX3X to the nuclear envelope.** We previously reported that DDX3X is present in mature HSV-1 virions and is needed for the optimal propagation of HSV-1, as depletion of DDX3X reduces extracellular viral yields by 50 to 70%. This reduction was initially attributed to its ability to modulate viral gene expression, which is anticipated for an RNA helicase[21,23,25]. To define where the virus first encounters DDX3X and might incorporate it, we examined the localisation of DDX3X during an HSV infection using our standard 17[+] lab strain. As expected[75], the nucleo-cytoplasmic shuttling DDX3X was largely cytoplasmic and diffuse in uninfected cells with few nuclear speckles (Fig. 1A; mock). However, DDX3X was relocalised to aggregates near the nucleus early during the infection (Fig. 1A; 3 h post-infection (hpi)), consistent with its ability to form stress granules under certain conditions[76]. At 6 hpi, most of these initial aggregates dissipated and were progressively replaced by a nuclear rim staining containing speckles. This was particularly obvious by 9 hpi and present in 80–85% of the cells (Fig. 1a), a time when most of the newly produced HSV-1 capsids exit the nucleus. To quantify this relocalisation, we labelled cells with antibodies targeting the lamin-B receptor (LBR), an inner nuclear membrane marker. While DDX3X was detected at that membrane in 15.6% ± 8.6 of uninfected cells, that ratio doubled to 36.4% ± 3.9 in infected cells with a $p$-value of 0.0367 (Fig. 1b). HSV-1 thus clearly redirected some of the DDX3X to the nuclear envelope late in infection.

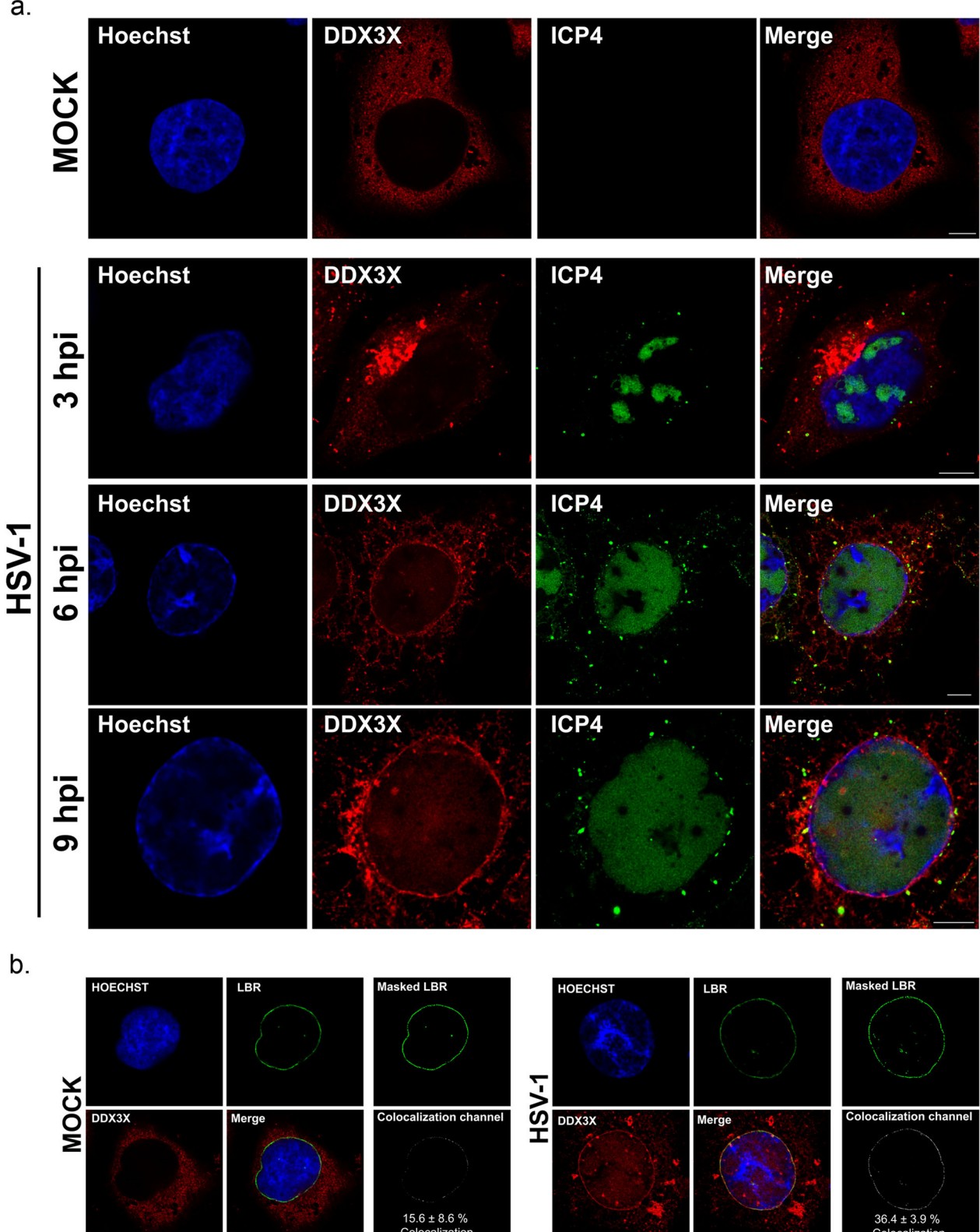

**Fig. 1 HSV-1 redirects DDX3X to the nuclear rim at late time points. a** HeLa cells grown on coverslips were either mock treated or infected with wild-type strain 17$^+$ virus at an MOI of 5. The infection was stopped at 3, 6 or 9 h post-infection (hpi). Cells were fixed and processed for immunofluorescence (DDX3X: red; ICP4: green; Hoechst 33342: blue). Samples were imaged using STED microscopy. Scale bars represent 5 μm. The images represent three independent experiments. **b** Quantification of DDX3X relocalisation to the nucleus was performed by seeding HeLa cells on coverslips and mock treated or infected with wild-type strain 17$^+$ virus at an MOI of 5. At 9 hpi, cells were fixed and processed for dual colour STED microscopy using anti-DDX3X (red) and anti-LBR (green) antibodies. In three independent experiments, twelve random whole cells (z-stacks) were imaged for both conditions. Next, images were processed with the Imaris software to calculate the amount of DDX3X found on a masked LBR channel. Scale bar represents 5 μm. ±Standard deviation of the means (bilateral Student *T*-tests; $p = 0.0367$).

**DDX3X modulates perinuclear virion accumulation**. Given the repositioning of DDX3X, we wondered if this was functionally relevant for the virus. We previously reported that DDX3X is required for the optimal expression of viral genes with downstream consequences on viral genome copies, capsid assembly and infectious particle abundance[25]. To probe if the helicase additionally impacts nuclear egress, we performed a detailed analysis of the HSV-1 egress pathway by electron microscopy in the context of RNA interference targeting DDX3X, using the above 17[+] viral strain. We also relied on previous siRNA reagents previously shown to be efficient at reducing DDX3X levels in cells (70% inhibition at the protein level) and inhibiting DDX3X incorporation in mature extracellular virions (>90% reduction) without any significant impact on cell viability (80% cell viability)[23,25]. As anticipated, all viral intermediates, including A-, B- and C-nuclear capsids, perinuclear virions and cytoplasmic capsids as well as mature extracellular virions were detectable during a wild-type infection in the presence of normal levels of DDX3X (Fig. 2a). Most interestingly, depletion of DDX3X led to reduced quantities of viral particles at the plasma membrane and an accumulation of wild-type enveloped viral particles in nuclear herniations (Fig. 2b). While there were no differences in the relative abundance of nuclear capsids (see %nuclear capsids) upon DDX3X depletion, there was a statistically significant accumulation of mature C-capsids (twofold) and perinuclear virions (sixfold) at the expense of extracellular viral particles when considering DNA-filled viral particles (Fig. 2c). Thus, DDX3X impacted the accumulation of perinuclear virions. This phenotype is reminiscent of that observed for $pU_S3$, which modulates NEC function during viral budding but leads to an accumulation of perinuclear virions in herniations when defective (see introduction). We also noted a slightly greater retention of C-capsids in the nucleus (twofold increase in DDX3X depleted cells) that also suggested a possible role for DDX3X in nuclear capsid budding, although this did not reach statistical significance. Overall, this indicated that DDX3X was required for the optimal release of HSV-1 capsids from the nucleus, an unforeseen role for a mammalian RNA helicase.

**DDX3X has multiple potential viral partners**. To decipher how DDX3X might alter viral nuclear egress, we hypothesized that DDX3X, directly or indirectly, interacts with at least one viral protein implicated in that pathway. To identify such partners, we developed a cell line stably expressing protein G tagged DDX3X to perform pull downs. This cell line exhibited a similar expression level and localisation as the endogenous DDX3X, as determined by Western blotting and immunofluorescence (Fig. S1). This was in line with our previous report that the tagged DDX3X behaves normally in both mock treated or HSV-1 infected HeLa cells[25]. We then infected the cells and collected cell lysates at either 4 or 12 hpi to distinguish early (no capsids made yet) and late (release of newly assembled viral particles) stages of the infection. Since DDX3X functionally interacts with an array of cellular proteins (uniprot.org), we used uninfected cells as control. Using triplicate independent mass spectrometry analyses and stringent conditions (95% peptide and protein confidence, 2 peptides minimum and detection in all 3 experiments), four viral proteins were uniquely identified at 4 hpi, another four at 12 hpi and seven interacted with DDX3X at both time points (Fig. S2a). About half of the targets were tegument proteins (40%), 27% nonstructural viral proteins, 20% envelope glycoproteins and 13% capsid proteins (Fig. S2b)[21]. These proteins were not simply viral particles that co-purified with DDX3X since many other virion components were absent in the samples, including the very abundant major capsid protein VP5.

**DDX3X forms a complex with the NEC**. Given the presence of DDX3X on nuclear membranes and its modulation of viral nuclear egress, the identification of the NEC component $pU_L31$ as a DDX3X binding partner was intriguing since the NEC is involved in capsid envelopment at the inner nuclear membrane. We therefore validated their interaction in reciprocal co-IP experiments. We resorted to a recombinant virus coding for a HA-tagged $pU_L31$ since the $pU_L31$ antibody performed poorly for IP. Note this mutant is in the HSV-1 F strain background. As controls, cells were mock treated or infected with wild-type strain F virus (no HA tag). We then purified $pU_L31$ using commercially available anti-HA agarose beads and examined the presence of DDX3X by Western blotting. The data confirmed the formation of a complex (DDX3X was co-IP by the HA antibody) and the specificity of the antibodies (DDX3X was only present in cells infected with the recombinant virus and neither $pU_L31$ nor HA antibodies detected these proteins in the mock or wild type infected cell lysates) (Fig. 3a). For the reverse condition, cells were infected with wild-type virus, endogenous DDX3X retrieved with the DDX3X R648 polyclonal antibody[77] and $pU_L31$ probed by Western blotting. As expected, $pU_L31$ was detected when DDX3X was enriched, but not in the absence of the primary antibody (Fig. 3b). To validate the interaction at the nuclear envelope, immunofluorescence studies were performed, in infected cells by labelling HA (for the HSV-1 $U_L31$-HA virus) and $pU_L34$ (as a NEC and nuclear membrane marker). The results confirmed the above findings in that DDX3X partially co-localises with $pU_L31$ at the nuclear envelope (Fig. S3). Furthermore, this further extended our above findings to the HSV-1 F strain.

We next mapped the DDX3X residues (see map in Fig. 3c) required to form a complex with $pU_L31$, taking advantage of a collection of bacterial constructs graciously provided by Dr. Arvind Patel and expressing various GST tagged DDX3X fragments[78]. As detailed in the Methods section and legend, infected cell lysates were passed over DDX3X fragments immobilized on Sepharose beads and analysed by Western blotting to see if $pU_L31$ binds to these fragments. Note that although the different fragments were expressed at similar levels (Fig. S4), they bound to glutathione Sepharose beads with different efficiency (Fig. 3d), presumably because of their distinctive 3D folding or solubility. To avoid any bias, we monitored DDX3X binding to the beads by incorporating 2,2,2-Trichloroethanol (TCE) in the SDS-PAGE gels, which enabled total protein imaging under UV illumination prior to their transfer onto PVDF membranes to detect $pU_L31$ by Western blotting[79,80]. This made it possible to estimate the $pU_L31$ to DDX3X or control GST signal ratios within the same gels (Fig. 3e). These experiments revealed that $pU_L31$ exclusively bound to the DDX3X carboxyl terminus, predominantly through the 498-662 fragment (Fig. 3d-e). Though this binding could be indirect as we used infected cell lysates as a source of $pU_L31$, these experiments nonetheless confirmed a physical interaction between DDX3X and the NEC.

**DDX3X nuclear membrane recruitment is NEC dependent**. We wondered what might drive DDX3X to the nuclear membrane during an HSV-1 infection. We surmised that $pU_L31$ may capture DDX3X at the nuclear membranes and evaluated if the absence of $pU_L31$ would alter the intracellular localisation of DDX3X. Indeed, an infection with a $pU_L31$ deletion mutant virus recruited significantly less DDX3X to the nuclear membranes (Fig. S5). Since the soluble $pU_L31$ is peripherally associated with the inner nuclear membrane through its interaction with the integral protein $pU_L34$[43], a mutant depleted for $pU_L34$ not surprisingly yielded the same phenotype with respect to DDX3X localisation (Fig. S5). It should be noted that $pU_L34$ localisation was somewhat perturbed by the absence of $pU_L31$. Hence $pU_L34$ was partly nuclear membrane

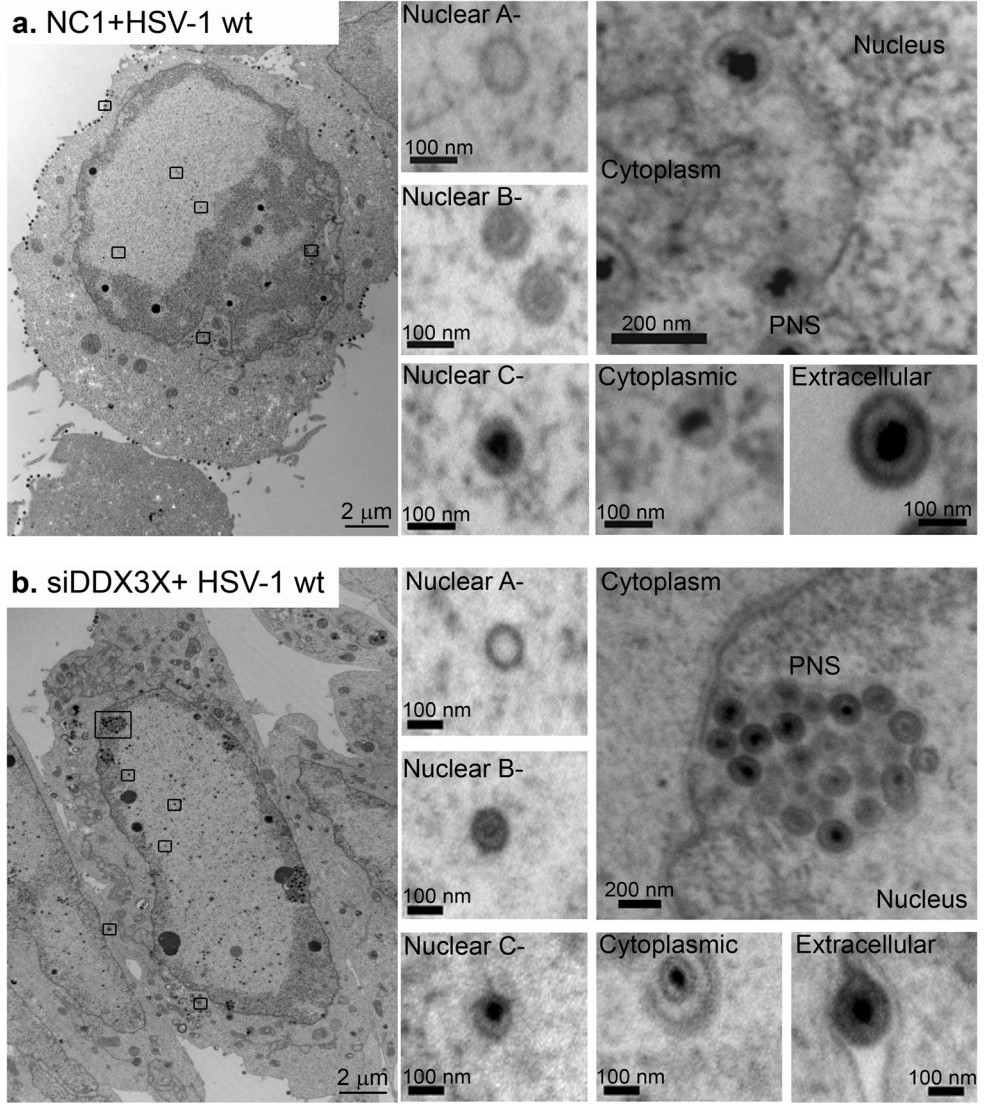

**c.**

| Cell Fraction | Particle Type | NC1+WT HSV-1 | | | | siDDX3X+WT HSV-1 | | | |
|---|---|---|---|---|---|---|---|---|---|
| | | Total Count | | Nuclear Capsids (%) | DNA-Filled Particles (%) | Total Count | | Nuclear Capsids (%) | DNA-Filled Particles (%) |
| | | N1 | N2 | | | N1 | N2 | | |
| **Nucleus** | A-Capsids | 73 | 189 | 12.9±1.6 | - | 75 | 130 | 9.3±0.3 | - |
| | B-Capsids | 381 | 1008 | 68.2±9.5 | - | 574 | 1112 | 75.8±7.2 | - |
| | C-Capsids | 195 | 99 | 18.8±11.2 | 8.7±0.6 | 188 | 96 | 14.8±7.6 | 18.5±0.4* (p=0.0108) |
| **PNS** | Mature | 83 | 135 | - | 7.5±3.5 | 384 | 250 | - | 43.0±4.2* (p=0.0252) |
| **Cytoplasm** | Mature | 961 | 467 | - | 42.1±4.0 | 280 | 132 | - | 26.6±1.6 |
| **Extracellular** | Mature | 840 | 525 | - | 41.6±1.2 | 138 | 51 | - | 11.8±2.1* (p=0.0153) |

**Fig. 2 DDX3X modulates HSV-1 exit from the perinuclear space.** HeLa cells were seeded on 6-well plates and treated with **a** control non-targeting NC1 or **b** siDDX3X for 48 h then infected with wild-type strain 17⁺ virus at an MOI of 5. At 12 hpi, cells were fixed and prepared for electron microscopy (see Methods). The images show whole cell views with zooms on A-, B- and C-nuclear capsids as well as viral particles in the cytoplasm and the cell surface. **c** Fourteen cells were randomly selected from each infected condition in two independent experiments (N1; N2). A-, B- and C-nuclear capsids as well as viral particles in perinuclear space (PNS), cytoplasm and extracellular space were counted. The percentages (±standard error of the means) refer to the relative amounts of nuclear capsids (% Nuclear capsids) or the total viral particles containing the viral genome (% DNA-filled particles, i.e., C-capsids, perinuclear virions, cytoplasmic viral particles, and extracellular virus). Bilateral Student *T*-tests were performed. *p < 0.05 and **p < 0.01.

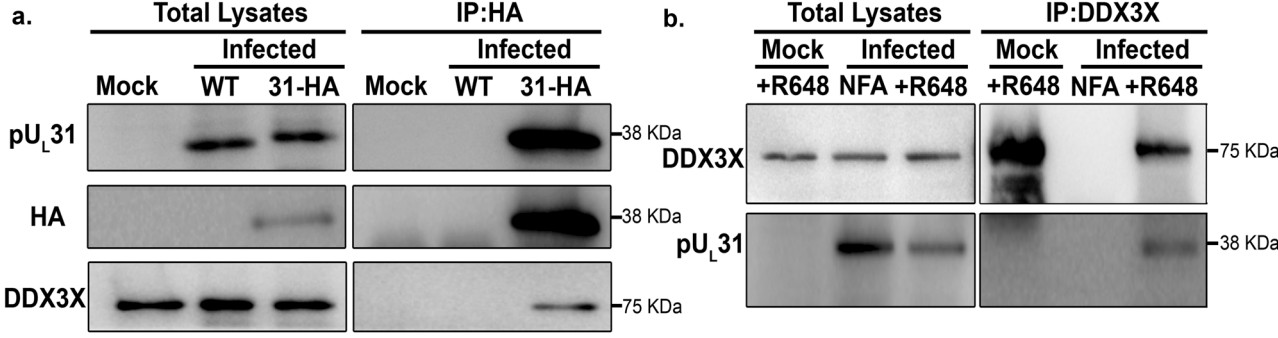

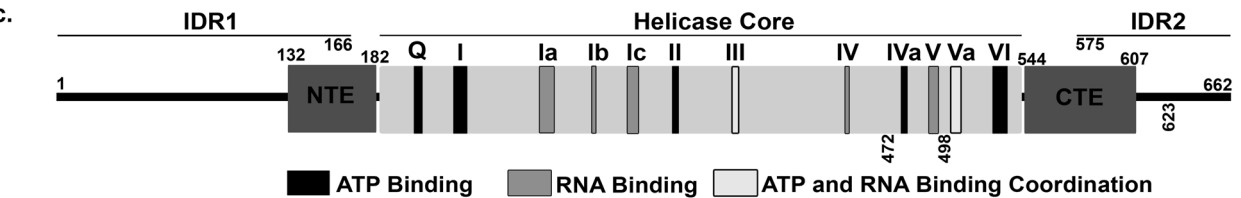

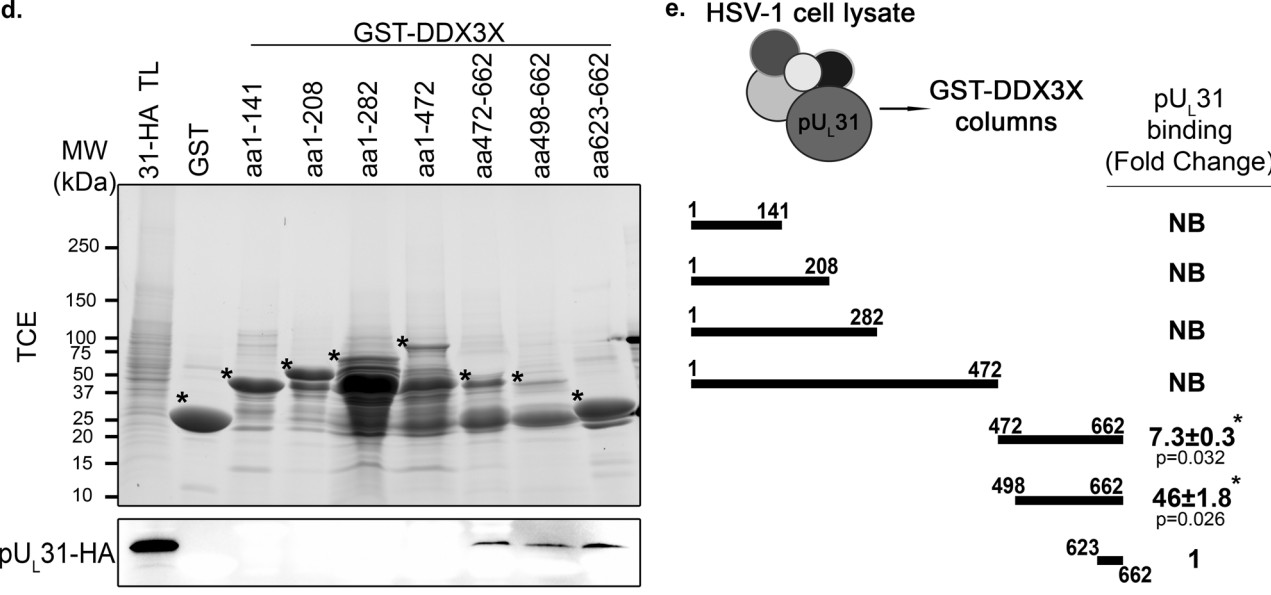

**Fig. 3 The DDX3X carboxyl tail DDX3X physically interacts with the HSV-1 nuclear egress protein pUL31.** The interaction of DDX3X and pUL31 was assessed using co-IP. **a** HeLa cells were either mock-treated or infected with wild-type strain F virus (negative controls) or with HA-tagged pUL31 virus at an MOI of 5 for 12 hpi. Cells were lysed and HA-tagged pUL31 pulled down using anti-HA agarose beads. Total lysates and eluates were analyzed by WB by probing pUL31 (anti HA or pUL31) and DDX3X (anti-human DDX3 rabbit R648 polyclonal serum) as indicated to the left of the blots. **b** The reverse co-IP was also performed with HeLa cells either mock-treated or infected as above but by precipitating DDX3X (anti-human DDX3 rabbit R648 polyclonal serum and protein A agarose beads). The presence of DDX3X and pUL31 in the eluates were assessed by Western blot using anti-DDX3X R648 or anti-pUL31 as shown to the left of those blots. **c** Schematic map of DDX3X showing its different domains and motifs (adapted from[103]). IDR, NTE and CTE represent intrinsically disordered domains, N- and C-terminal extensions respectively. The helicase core (residues 182-544) represents the minimally active portion of DDX3X. All conserved DDX3X motifs are colour coded as defined by the legend on the figure. These motifs modulate ATP and RNA binding or the coordination between the two functions. **d** Recombinant GST-tagged DDX3X constructs or a GST alone control were captured on glutathione Sepharose 4B beads. HeLa infected lysates (HA-tagged pUL31 virus, MOI of 5, 24 hpi) were pre-cleared on the GST preloaded glutathione Sepharose beads then incubated with the various enriched GST-DDX3X constructs for 3 h. The beads were boiled in sample buffer to release the bait. Samples were analyzed on 5–20% gradient SDS-PAGE containing 0.5% TCE. The level of the bead bound GST-tagged constructs was analyzed by UV exposure (TCE panel) on a ChemiDoc (BioRad). Binding of pUL31 was evaluated by Western blotting using a HA-tag antibody (bottom panel). Stars highlight the position of each GST construct. TL represents total lysates prior to its loading on the beads. **e** Quantification of pUL31 binding to the beads for each DDX3X construct (drawn schematically). Binding was calculated by normalizing the HA-tagged pUL31 Western blot signal to the TCE signal for each GST chimera. Fold changes were measured by arbitrarily setting the binding to GST-DDX3X aa 623-662 to one. Absence of binding is defined as NB (no binding). ±Standard error of the means (bilateral Student *T*-tests; *p < 0.05). The figure represents four independent experiments.

bound but spread over an ER-like compartment in the absence of pU$_L$31, suggesting some erroneous membrane targeting, a phenotype previously reported by Roller and colleagues[43]. Given the role of pU$_S$3 on the NEC nuclear egress machinery, we also probed its impact on DDX3X nuclear localisation. Interestingly, a U$_S$3 deletion viral mutant similarly recruited DDX3X less efficiently to the nuclear membrane (Fig. S5), suggesting that phosphorylation of NEC components or their smooth positioning in the nuclear membrane may contribute to DDX3X recruitment onto the nuclear membranes. In contrast, depletion of cell-associated DDX3X by RNA interference had no impact on the nuclear localisation of the NEC components (Fig. S6). These findings were thus most consistent with a model whereby pU$_L$31 captures DDX3X at the nuclear envelope.

**DDX3X promotes the formation of large VP5 foci at the nuclear periphery.** While the above EM study highlighted the role of DDX3X during the de-envelopment stage of the nuclear viral capsids, the possible accumulation of mature capsids within the nucleus was puzzling. To evaluate in detail how the physical interaction between the NEC and DDX3X impacts viral nuclear egress, we monitored by high-resolution STED confocal microscopy both DDX3X and the viral capsids, as detected with antibodies targeting the main capsid component VP5. As expected, DDX3X was essentially cytoplasmic in uninfected cells and partially redirected to the nucleus by the virus late in infection (Fig. 4a,b). Unexpectedly, DDX3X co-localised with VP5 in large slightly elongated capsid foci at the nuclear periphery (>1 μm and up to 4 μm) but not with more diffuse or smaller round VP5 foci (0.5–1 μm). Quantification of the data from six cells in each of the three independent experiments revealed that 100% of these large VP5 foci were DDX3X positive. These large capsid foci, visible in ~20% of the infected cells, did not co-localise with ICP4 (Fig. 4c), a viral marker of replication compartments[81], but did occasionally stain positive for the viral envelope protein gB, suggesting the capsids were closely apposed to the nuclear envelope or perhaps represented perinuclear enveloped virions (Fig. 4d).

To evaluate if the large capsid foci were DDX3X-dependent, we repeated the experiments in the presence or absence of the host protein. In infected cells treated with non-targeting NC1 siRNA, we observed and quantified four different VP5 distribution patterns (diffused nuclear, nuclear and cytoplasmic, small nuclear foci and the aforementioned large nuclear foci), all best visible in rendered images (Fig. 5a). In DDX3X depleted cells, the VP5 staining pattern was generally more diffuse with the statistically significant loss of most small foci and the absence of the large elongated foci (Fig. 5a, b). DDX3X thus seemed important for the generation of the capsid aggregates. Given the proximity of these large capsid aggregates with the nuclear periphery, we wondered if DDX3X alters the intracellular distribution of the VP5 foci by measuring their distance from the nuclear periphery. However, that analysis revealed that DDX3X did not alter the positioning of the VP5 foci (Fig. 5c). Interestingly, the bulk of the large VP5 aggregates was in close proximity of the nuclear envelope with 20–25% of all aggregates near the 0 μm axis (see large arrow in Fig. 5c). We concluded that DDX3X was required for the genesis of the larger aggregates at the nuclear periphery, but not their transport there from another location.

**DDX3X modulates C-capsid budding.** The DDX3X-dependent formation of large capsid foci at the nuclear periphery distinct from the viral replication compartments raised the question as to whether these capsids are those exiting the nucleus. This is not trivial as the HSV-1 generates four different types of nuclear capsids (see introduction). To explore if these large capsid aggregates were C-capsids, we compared the VP5 expression pattern formed by wild-type virus with those generated by the

viral mutant vFH421 coding for HSV-1 Δ1-50 U$_L$25, which produces highly reduced C-capsid levels but still efficiently makes other capsid types[34,36,82,83]. Note that these mutants are based on the KOS strain, so we compared them to the corresponding wild-type parental strain. Figure 6a (top panels) shows once again the usual distribution of the four VP5 expression profiles in wild-type infected cells, indicating that these phenotypes also extent to a third HSV-1 strain. However, in cells infected with the viral mutant, both small and large VP5 nuclear foci were statistically reduced with only 5% of the cells exhibiting large foci where both VP5 and DDX3X colocalised (Fig. 6a, b). While some residual foci persisted, we presume they may represent the limited A- and B- nuclear capsids that do escape the nucleus[29]. Interestingly, pU$_L$34 localisation was strongly perturbed by the absence of nuclear capsids and displayed an intranuclear staining. While unexpected for a transmembrane protein, pU$_L$34 was also found in a proteomics study of purified HSV-1 nuclear capsids (i.e. using MS rather than antibodies), thereby orthogonally confirming this odd observation that we cannot explain at the moment[36]. Nonetheless, this overall suggests that the large VP5 foci were both DDX3X and C-capsid dependent, consistent once again with a role for DDX3X during HSV-1 nuclear egress.

To better understand the mechanism by which DDX3X modulates HSV-1 nuclear capsid egress, we next evaluated if C-capsids were needed for DDX3X nuclear localisation. This was evaluated in the context of mutant viruses that fail to assemble viral capsids altogether (mutant lacking the major capsid protein VP5[84]) or only assemble A- and B- but not C-nuclear capsids (Δ1-50 U$_L$25). Figure 7 shows that the ΔVP5 viral mutants did not redirect pU$_L$34 to the nuclear envelope. In that context, DDX3X was also, not surprisingly, absent from the nuclear envelope since the NEC recruits it there (Fig. 7). Most interestingly, these phenotypes were also found in cells infected with the C-capsid deficient virus (Δ1-50 U$_L$25), indicating these phenotypes were C-capsid driven.

An interesting aspect was whether the large VP5 foci seen above precedes capsid association with the NEC. To test this, we resorted to confocal microscopy and Z stacks in the context of U$_L$31, U$_L$34 or U$_S$3 deficient viral mutants. Quantification of those results indicated that while large VP5 nuclear foci were visible as before in roughly 20% of wild-type infected cells, that proportion was largely and statistically significantly reduced upon NEC disruption (Fig. S7 and Fig. 8a). Thus, formation of the large elongated capsid foci was hampered in the absence of the NEC, suggesting the capsids first encountered the NEC then agglomerated. To more directly probe this phenotype, infected cells were reimaged by electron microscopy. Upon close examination of wild-type infected cells, the only elongated viral aggregates >1 μm at the nuclear periphery were enveloped C-capsids within the perinuclear space (Fig. 8b). Interestingly, the presence of these capsid foci within the perinuclear space readily explained their elongated nature, being trapped between the two nuclear membranes. Overall, these experiments revealed that DDX3X also plays a role at the nuclear envelope during HSV-1 perinuclear virion morphogenesis, i.e., during capsid budding at the inner nuclear membrane, in addition to its implication during the subsequent capsid de-envelopment at the outer nuclear membrane.

**DDX3X is absent on purified nuclear capsids.** During capsid budding at the inner nuclear membrane, one appealing scenario is DDX3X may bridge C-capsids to the nuclear membrane by binding a viral component on the capsid. This could either be pU$_L$31, which is present on nuclear capsids[36,54,55], or one of the structural viral proteins identified in the present study as DDX3X binding partners (Fig S2), several of which were reported on nuclear capsids[36,85,86]. To probe this option directly, total capsids

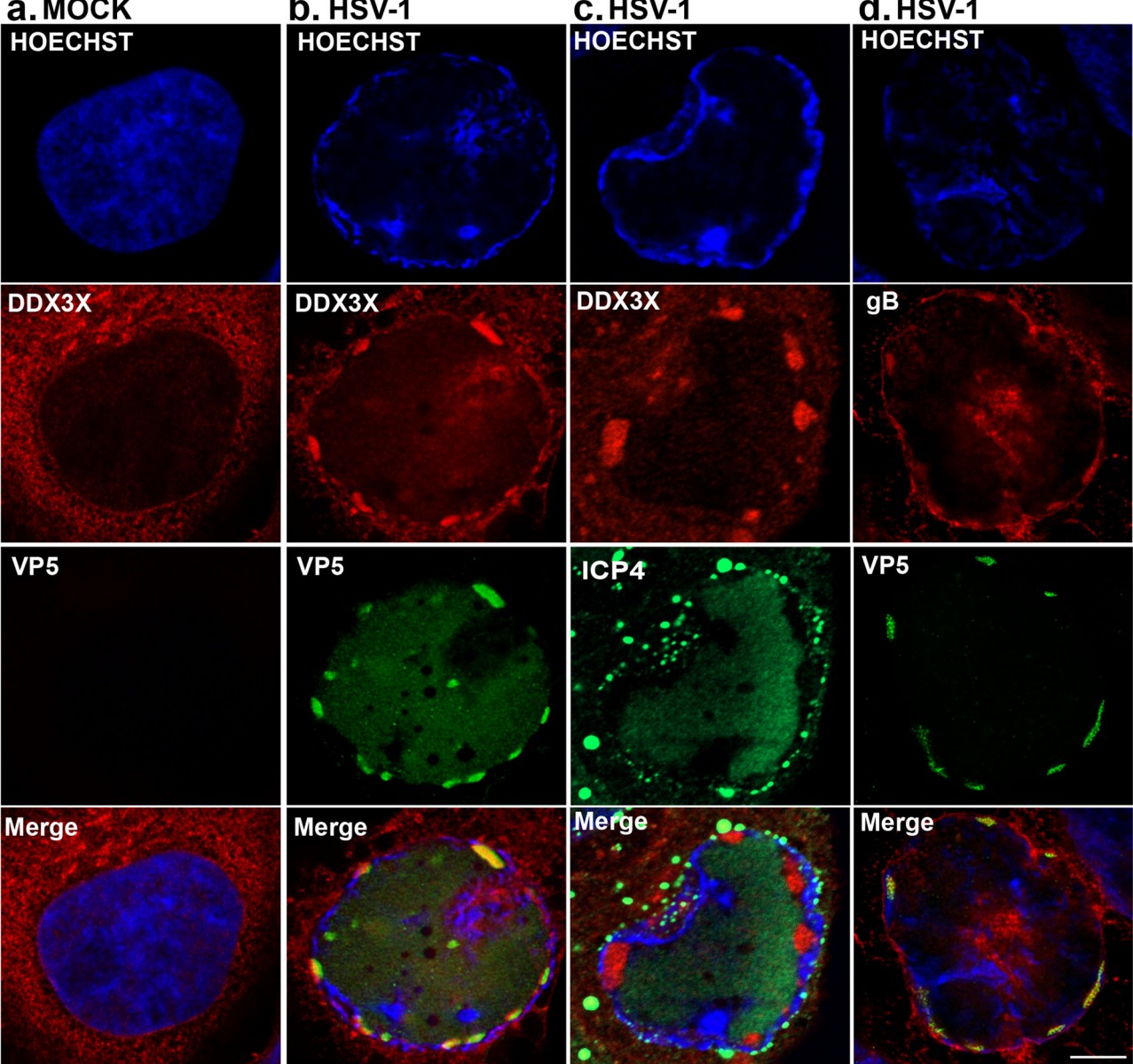

**Fig. 4 DDX3X colocalises with large VP5 aggregates at the nuclear rim distinct from the viral replication compartments.** HeLa cells seeded on coverslips were either **a** mock-treated or **b–d** infected wild-type strain 17⁺ virus at an MOI of 5 for 9 hpi. Cells were fixed and labelled for **a, b** DDX3X (red) and VP5 (green) **c** DDX3X (red) and ICP4 (green) or **d** gB (red) and VP5 (green). All nuclei were stained with Hoechst 33342 (blue). Cells were imaged using dual colour high-resolution STED microscopy. Scale bar represents 5 μm. The images are representative of three independent experiments.

were purified from infected nuclei under physiologic salt conditions that maintains tegument proteins on the capsids[86]. As positive control, we also monitored extracellular virions, which were previously shown to incorporate DDX3X[23]. Western blotting indeed confirmed the presence of DDX3X in extracellular virions but not on the nuclear capsids (Fig. S8). This was consistent with our recent mass spectrometry study that did not reveal the presence of DDX3X on purified A-, B- or C-nuclear capsids[36]. While we cannot exclude that DDX3X fell off the capsids during their purification, it seems more likely that the protein is not recruited to the capsids prior to their interaction with the NEC and that DDX3X rather interacts with incoming C-capsids at the nuclear membrane.

**DDX3X promotes pU_S3 incorporation into the viral particles.** Thus far, DDX3X seems to promote the passage of the newly

assembled viral capsids through both the inner and outer nuclear membranes. In the former case, we showed above that DDX3X stimulates C-capsid budding by interacting with the NEC on the nuclear envelope, leading to the formation of perinuclear enveloped virions. However, how DDX3X might also promote perinuclear virion accumulation was puzzling. We hypothesized that DDX3X might recruit a key protein onto the viral particles during their budding that is needed for the subsequent stage of the infection. Of interest, newly assembled HSV-1 capsids mature by sequentially recruiting various tegument proteins within the nucleus, in the cytoplasm or the final envelopment site[38]. To decipher how DDX3X might promote the fusion of perinuclear virions with the outer nuclear membrane, we considered probing the tegument content of perinuclear virions, but this would require their purification, a difficult task only previously achieved by one lab[87]. Since purity remained an issue that cannot readily

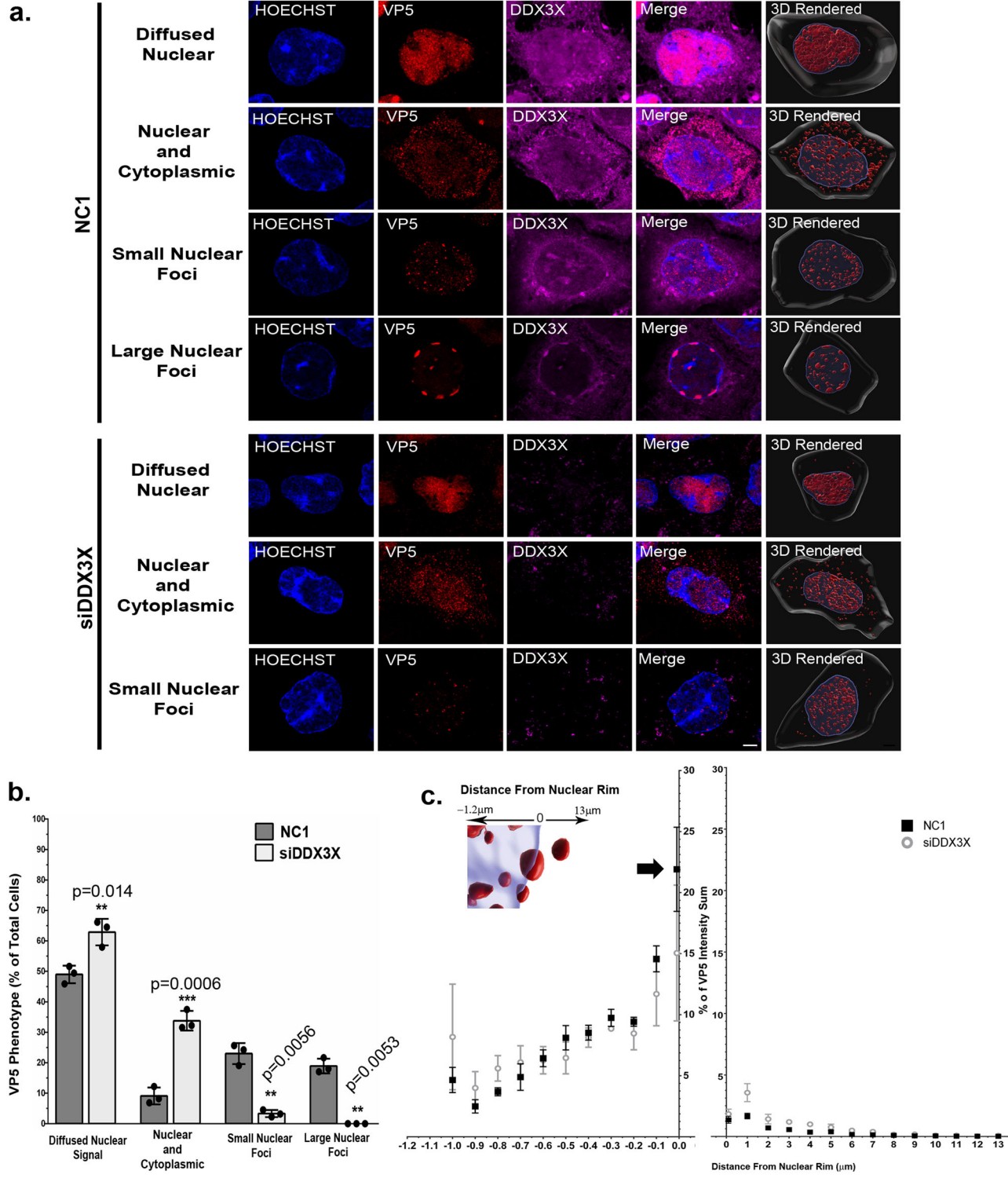

**Fig. 5 VP5 fails to concentrate at the nuclear rim in the absence of DDX3X. a** HeLa cells grown on coverslips were transfected with 25 nM of siDDX3X or non-targeting control siRNA (NC1) for 48 h. Cells were then infected wild-type strain 17⁺ virus at an MOI of 5 for 9 hpi. Fixed cells were labelled for DDX3X (magenta) and VP5 (red) and the nuclei stained using Hoechst 33342 (blue). This panel shows representative images of the different VP5 phenotypes observed and their 3D rendered illustrations (see Methods). The rendered images (Z-stacks acquired on a conventional confocal microscope). Scale bar represents 5 μm. **b** Eight random images (containing a total of about 200 cells) were analyzed for each condition to quantify the above VP5 phenotype. The bars indicate means and the standard error of the means. Bilateral Student $T$-tests, $**p < 0.01$; $***p < 0.001$; NS: not significant. **c** Twenty random cells per phenotype (six cells for the less abundant small nuclear foci in siDDX3X treated cells) were analyzed on the Imaris software to compare the distribution of the VP5 signals (% intensity sum) in relation to their distance from the nuclear rim (μM). Each point represents the mean of three independent experiments and error bars indicate the standard deviation of the means. So statistical differences were found in **c**.

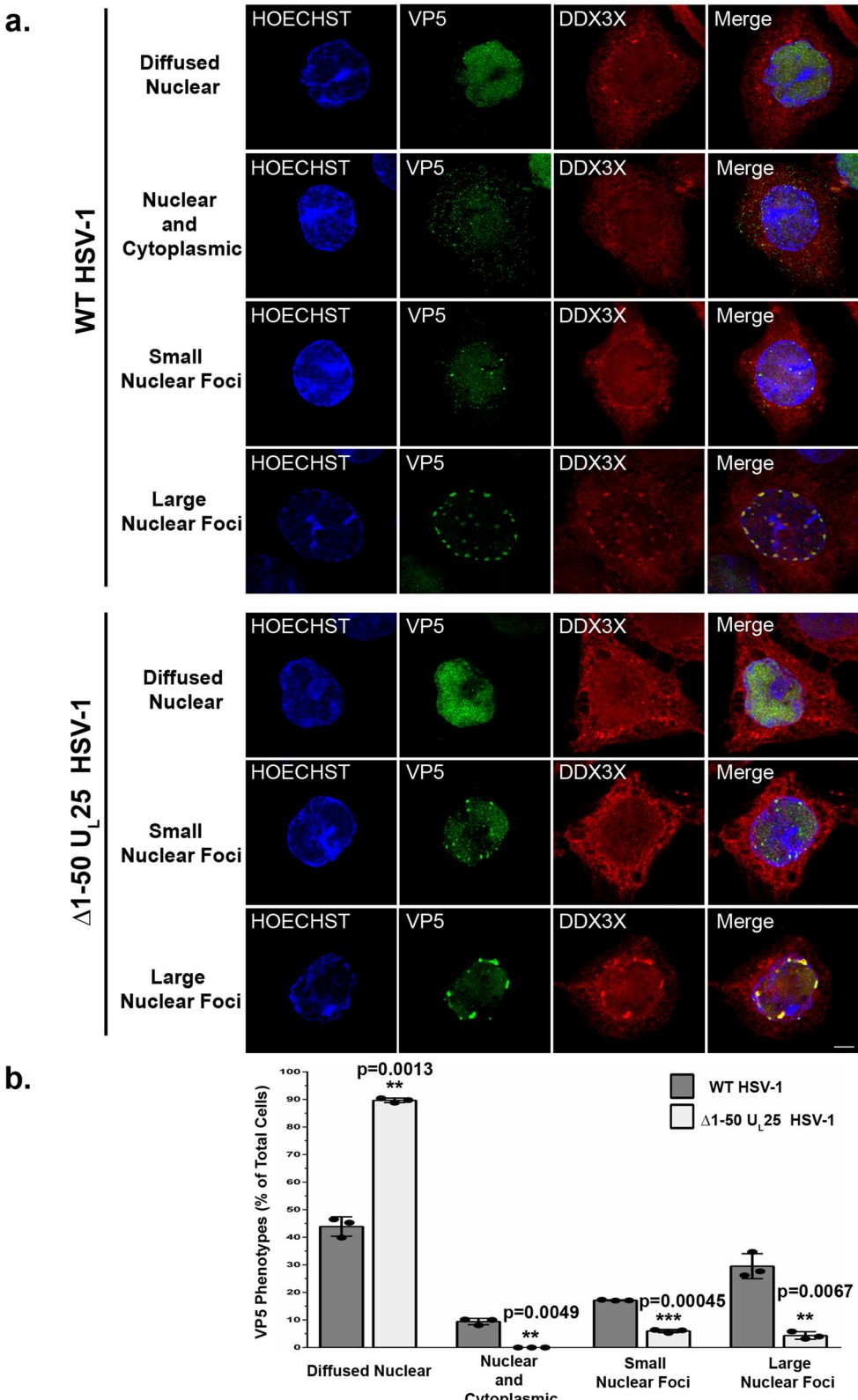

**Fig. 6 DDX3X and the large VP5 nuclear aggregates are C-capsids dependent. a** HeLa cells were mock-treated or infected at an MOI of 5 with the viral mutant Δ1-50 $U_L$25 (only makes A and B nuclear capsids) or corresponding KOS wild-type virus. At 9 hpi, cells were fixed and processed for immunofluorescence (DDX3X: red; VP5: green; Hoechst 33342 stained nuclei: blue). Images were acquired on a conventional confocal microscope. The scale bar represents 5 μm. **b** Ten random images (~200 cells per condition) were used to quantify the different VP5 phenotypes. The percentages are means of three independent experiments along with their standard deviation (bilateral Student *T*-tests; ***$p < 0.001$).

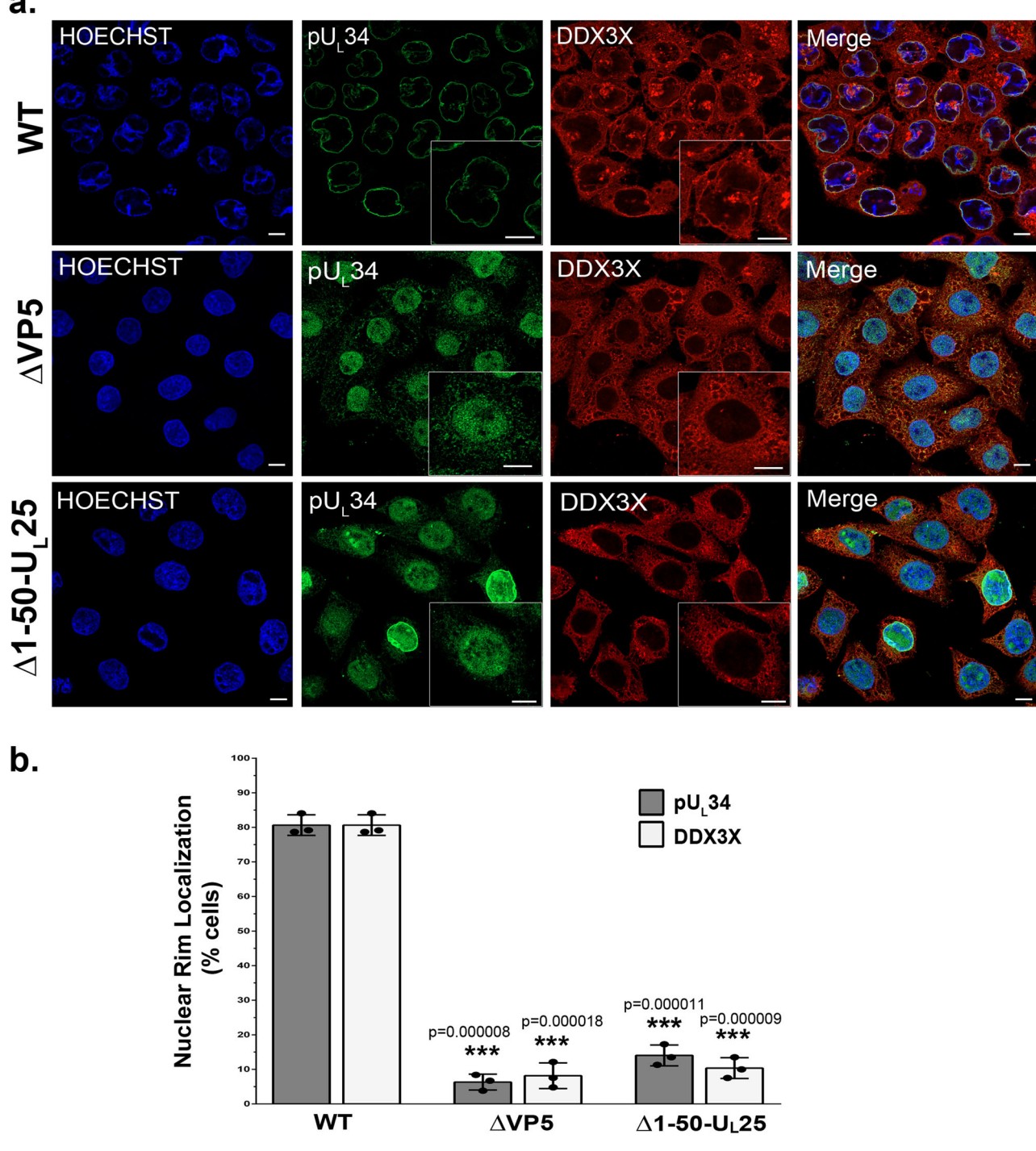

**Fig. 7 The presence of DDX3X on nuclear membranes is C-capsid dependent. a** HeLa cells seeded on coverslips were either mock-treated or infected at an MOI of 5 with the ΔVP5 or Δ1-50 U$_L$25 mutant viruses or the corresponding KOS wild-type virus. The cells were fixed at 9 hpi and labelled for DDX3X (red), pU$_L$34 (green) and Hoechst 33342 stained nuclei (blue). Zoom boxes were added to highlight DDX3X and pU$_L$34 localisation in each condition. All scale bars represent 5 μm. **b** Quantification of the nuclear pU$_L$34 and DDX3X using 100 random cells per condition. Bars indicate means and the standard deviation of the means (bilateral Student $T$-tests; ***$p < 0.001$). The data represents three independent experiments.

be addressed, we instead monitored the tegument composition of the end product of viral egress, namely mature extracellular mature virions produced by DDX3X depleted cells. As pU$_L$31 and pU$_L$34 are not components of mature extracellular virions, they were omitted from that analysis[21,45]. Of the seven tegument proteins tested by Western blotting, most showed no significant differences between viruses produced in cells treated with non-targeting control siRNA or those from siDDX3X treated cells

(VP1/2 (pU$_L$36), ICP4, ICP0, VP13/14 (pU$_L$47), VP16 (pU$_L$48) and VP22 (pU$_L$49)). However, a statistically significant decrease was found for pU$_S$3 upon DDX3X depletion (Fig. 9). This was quite exciting and highly relevant given the dual role of pU$_S$3 during HSV-1 egress through the two nuclear membranes (localisation of the NEC and fusion of the perinuclear virions with the outer nuclear membrane). We therefore probed if pU$_S$3 interacts with DDX3X but perhaps eluded detection in our

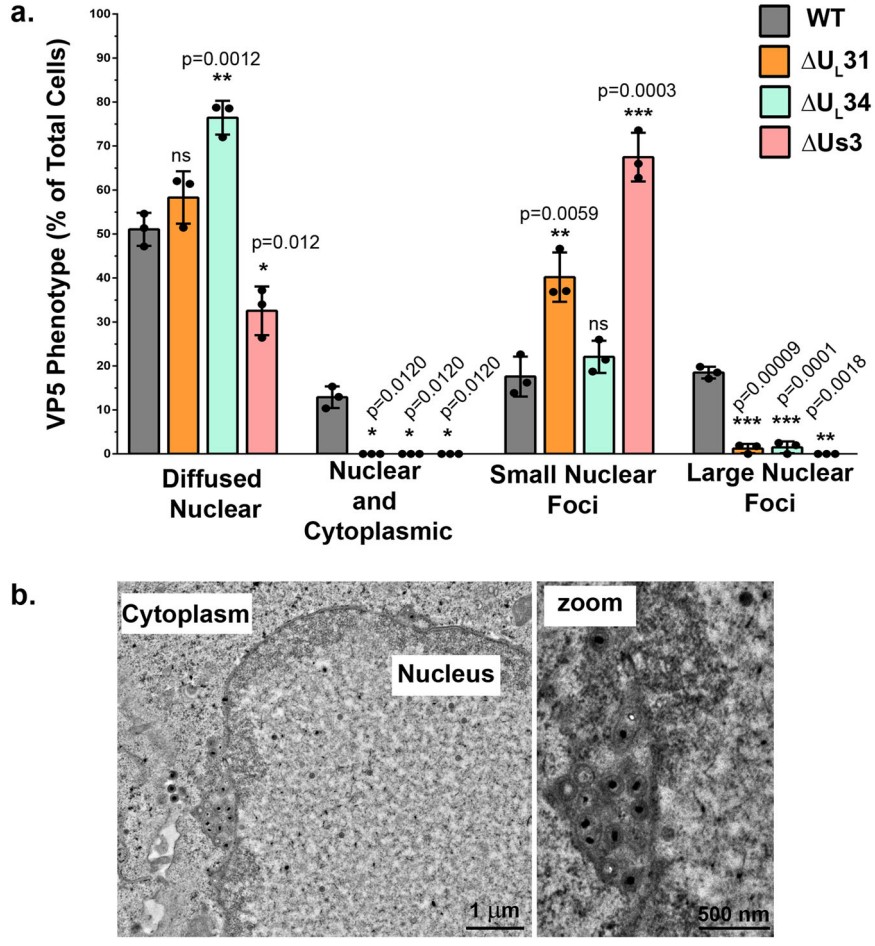

**Fig. 8 Large elongated VP5 foci at the nuclear periphery are enveloped C-capsids. a** HeLa cells grown on coverslips prior to infection. Twenty-four hours later, cells were infected with the ΔUs3, ΔUL31 or ΔUL34 mutant viruses or corresponding wild-type virus (F strain) at an MOI of 5. At 9 hpi the cells were treated for conventional confocal microscopy and labelled for DDX3X, VP5 and pUL34 (see Fig. S7 for IF images). At least 100 cells per condition were analyzed to quantify the intracellular VP5 distribution. The percentages are averages and standard deviations of the means of three individual experiments. Bilateral Student *T*-tests were performed and compared to wild-type virus (*$p < 0.05$, **$p < 0.01$ and ***$p < 0.001$). **b** HeLa cells were infected with wild-type strain 17+ virus at an MOI of 5 for 9 h. Next, cells were fixed and treated for electron microscopy. Several cells were imaged using by transmission electron microscopy. The images are representative of three independent experiments.

proteomics approach (Fig. S2). Initial experiments using total cell lysates failed to detect such interaction as per our MS experiments (data now shown). However, since both the soluble pUs3 and DDX3X proteins are present at the nuclear membrane ([58] and this study), we next biochemically purified nuclear membranes. While imperfect, the data revealed an efficient separation of cytoplasmic and nuclear fractions as well as a strong enrichment of nuclear membranes based on their respective cellular markers (Fig. 10a). Most interestingly, immunoprecipitation of DDX3X brought down pUs3 from infected cells but only in the nuclear membrane fraction and at best only poorly in other fractions (Fig. 10b). These data infer that DDX3X promotes pUs3 recruitment onto the viral particles, presumably during capsid docking/budding where the viral kinase would enable the subsequent release of the perinuclear virions into the cytoplasm.

## Discussion

The RNA helicase DDX3X has primarily been associated with cellular RNA metabolism, the interferon response and modulation of RNA viruses. However, we and others have reported that DDX3X is incorporated into mature viral particles for hepatitis B virus, HSV-1, PRV and HCMV, all of which are DNA based

viruses[18,20–22,88]. We also previously published that DDX3X modulates HSV-1 gene expression with downstream effects on viral genome copy numbers, capsid assembly and infectious particles in an interferon-independent manner[23,25]. This extended the range of action of DDX3X to both RNA and DNA viruses. However, it was not clear how, where and why DNA viruses recruit the cellular helicase. The present study probed these critical aspects and reveals that HSV-1 encounters DDX3X at the nuclear envelope, where it promotes viral nuclear egress via its interactions with the viral nuclear egress complex and pUs3. Since DDX3X does not appear to be required to infect cells[25], the presence of DDX3X in mature virions is thus a remnant of its role during that step of the viral life cycle. This is a most interesting finding for an RNA helicase.

At the nuclear periphery, DDX3X promoted the accumulation of large VP5 foci in all three strains tested in this study (17+, F and KOS). Since these foci did not co-localise with ICP4, a classical marker of viral replication compartments[81], and only formed in the presence of C-nuclear capsids, we initially considered they might be mature DNA-containing capsids ready to exit the nucleus. However, confocal microscopy revealed these capsid foci partially co-localised with the envelope gB viral protein and required the NEC, suggesting they were either very

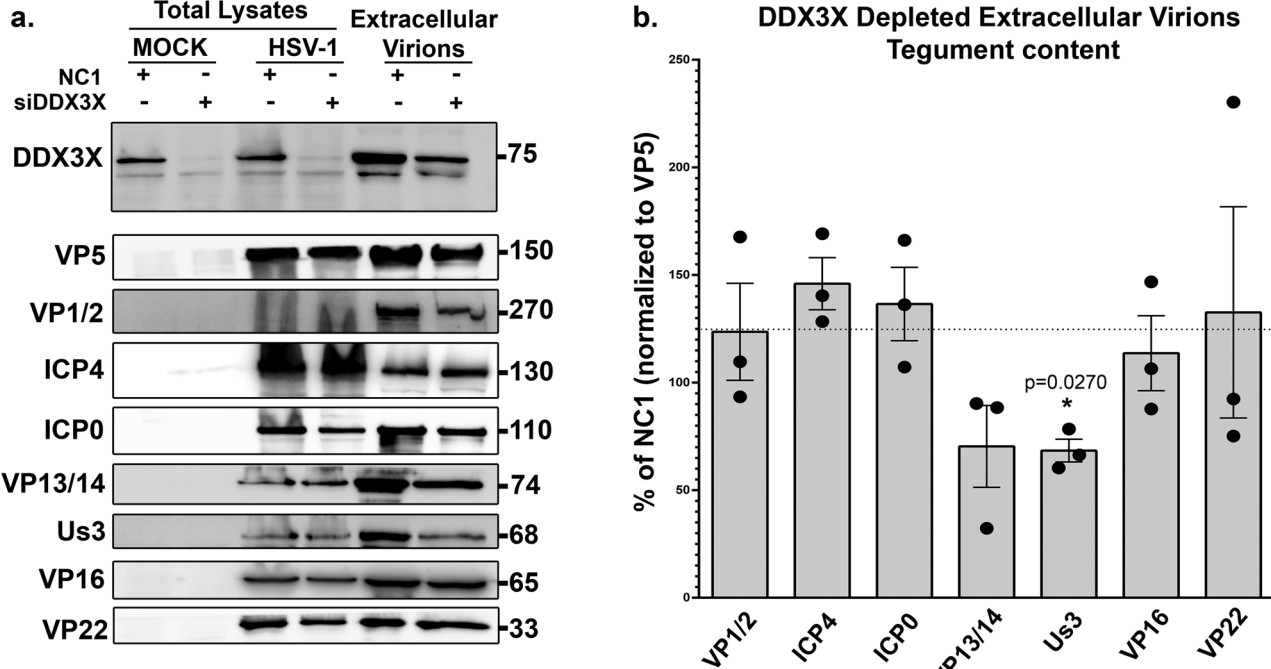

**Fig. 9 DDX3X promotes optimal pUs3 virion incorporation. a** HeLa cells were grown on 245 mm plates and treated with control non-targeting NC1 or siDDX3X for 48 h. Transfected cells were mock-treated or infected with wild-type strain 17+ virus. At 24 hpi, supernatant containing the mature virions were collected and concentrated at 60,000 x g. They were next analyzed on 5–20% gradient SDS-PAGE and Western blotting, using a panoply of antibodies targeting various viral tegument proteins. Total lysates of the same cells collected in lysis buffer and analyzed in parallel served as negative (mock cells) and positive (infected cells) antibody controls. DDX3X depletion in the cell and virions was confirmed by probing DDX3X in the same samples. **b** Image Lab version 5.0 was then used to quantify the Western blots (note that detection of the proteins was done on an instrument with a 4-log linear range, not on X-ray film). VP5, which served as a loading control, was used to normalize the data of the infected samples. The bars indicate means of three independent experiments and the standard error of the means (bilateral Student $T$-tests; *$p < 0.05$). pUs3 was the only statistically different tegument in the DDX3X depleted virions.

closely apposed to the nuclear membrane or had already reached the perinuclear space, the latter of which was ultimately confirmed by electron microscopy. These findings suggest that DDX3X is required for optimal capsid budding across the inner nuclear membrane.

A proteomics analysis to discover potential viral proteins driving DDX3X to the nuclear envelope and facilitating capsid budding across the inner nuclear membrane identified up to 15 different viral proteins. Among them was pU$_L$31, a component of the NEC known to modulate HSV-1 nuclear budding. Formation of a DDX3X-pU$_L$31 complex was orthogonally confirmed by reciprocal co-immunoprecipitation and fluorescence microscopy and mapped to the carboxyl tail of DDX3X. Moreover, DDX3X nuclear recruitment was dependent on the entire NEC machinery including the soluble pU$_L$31 protein, but also the pU$_L$34 transmembrane protein, which is essential to anchor pU$_L$31 to the nuclear envelope, as well as the pU$_S$3 viral kinase that impacts the nuclear localisation of the NEC. At this point, we cannot rule out an indirect binding of DDX3X to pU$_L$31, which may occur through pU$_L$34. The statistically significant reduction in DDX3X nuclear localisation observed in the absence of pU$_L$31, while pU$_L$34 remained nuclear bound, however favours some pU$_L$31 implication. The intriguing absence of pU$_L$34 in the present MS analysis of DDX3X partners, despite its usual binding to pU$_L$31, further corroborates this view. In contrast, that MS analysis identified pU$_L$47, a protein that interacts with the NEC and mediates HSV-1 nuclear egress[89]. It will be most interesting to resolve its potential role but also the importance of NEC phosphorylation on DDX3X nuclear binding. In contrast, NEC nuclear localisation was DDX3X independent, indicating the

NEC recruits DDX3X and not vice versa. Our data also revealed that DDX3X nuclear localisation and the selective release of C-capsids are functionally linked since interdependent. We considered whether DDX3X might bring the capsids to the NEC and nuclear membranes, a function that has interestingly been proposed for pU$_L$31[55], but ruled out this scenario since DDX3X is absent on nuclear capsids. DDX3X rather appears to bridge the outgoing capsids to the NEC directly on the nuclear membrane by virtue of its interactions with the NEC and the multiple capsid/tegument components identified by MS as DDX3X binding partners. As these are rarely seen transient viral intermediates, they rapidly reach the perinuclear space once membrane bound.

One unforeseen observation from this study has been that pU$_L$34 nuclear positioning depends on the presence of C-capsids. This is counterintuitive given that the sole overexpression of the NEC in transfected cells is sufficient to drive it onto nuclear membranes[43,46,90,91]. However, the situation is clearly more complex in infected cells as the nuclear positioning of the NEC is modulated by pU$_S$3, pU$_L$31 and is even cell type dependent in transfected cells[43,45,59,92]. Moreover, in transfected cells or in vitro assays using artificial membranes, the NEC stimulates uncontrolled vesicle formation, which is absent in infected cells[48–50]. It ensues that the NEC is normally tightly controlled during the course of an infection. In agreement with this view, it has recently been shown that capsid-associated pU$_L$25 is an important player that favours the pentamerization of the NEC, thereby activating its membrane deforming potential[35]. Banfield and colleagues also elegantly demonstrated that the viral tegument protein pU$_L$21 counteracts the kinase activity of pU$_S$3 by binding to protein phosphatase 1, which in turn activates the

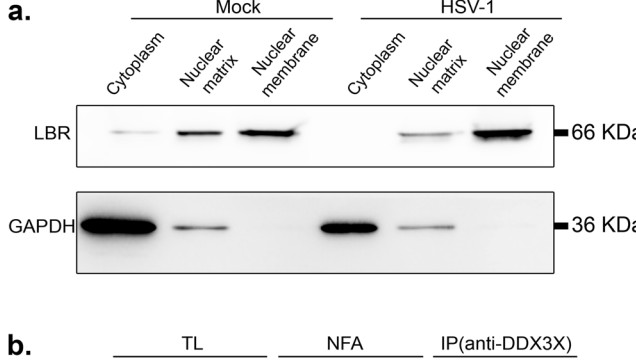

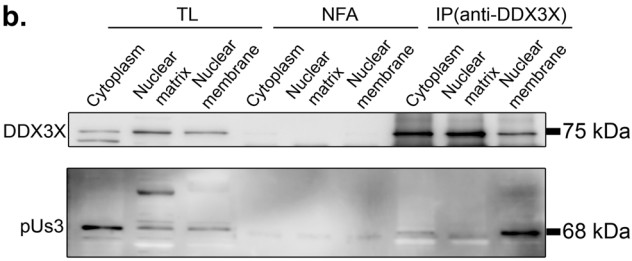

**Fig. 10 DDX3X physically interacts with pU$_S$3.** HeLa cells were either mock treated or infected with wild-type strain 17+ virus for 12 h. Cells were then fractionated into cytoplasm, nuclear matrix and nuclear membranes as detailed in Methods and **a** probed by WB for the GAPDH and LBR cytoplasmic and nuclear membrane markers respectively and **b** used to immunoprecipitate DDX3X using the R648 polyclonal serum and monitored for the presence of pU$_S$3. NFA: no first antibody. In parallel, total lysates were loaded on the gels. Results represent three independent experiments.

NEC[60,62]. They also reported that the tegument protein pU$_L$16 is implicated in the positioning of the NEC[93]. Thus NEC transfected cells are clearly lacking critical control components. In this light, it is relevant that pU$_L$34 localisation is dependent on the presence of nuclear capsids[51], which remains the best signal to turn on the NEC budding machinery. This is also consistent with cryo-EM data that show capsid-dependent NEC recruitment at the nuclear membrane[51].

A deeper analysis of the viral egress pathway revealed that DDX3X depletion led to the accumulation of perinuclear virions in nuclear herniations, i.e. invaginations, reminiscent of those seen for pU$_S$3 kinase dead or deletion mutants and some pU$_L$31 mutants[42,45,58–60]. How DDX3X also impacted this second step of HSV-1 nuclear egress was resolved by showing that DDX3X is required for the optimal recruitment of pU$_S$3 into mature virions and by providing evidence for their physically interaction on enriched nuclear membranes. This is consistent with previous reports that pU$_S$3 is absent from nuclear capsids but is recruited on nuclear membranes by the NEC and is present on perinuclear virions[36,45,58]. This highlights a mechanism whereby DDX3X facilitates the incorporation of pU$_S$3 onto the viral particles by interacting with it as the virus buds through the inner nuclear membrane, thereby promoting the subsequent pU$_S$3 mediated fusion of the perinuclear virions with the outer nuclear membrane, where pU$_S$3 is known to act. These findings highlight the coordinated passage of the viral capsids across the two nuclear membranes.

Overall, the present study highlights unforeseen roles for DDX3X in HSV-1 nuclear capsid egress through (1) its interaction with the viral nuclear egress machinery; (2) by stimulating the formation of large elongated C-capsids foci in the perinuclear space; (3) by stimulating pU$_S$3 recruitment onto the viral particles thus (4) promoting virion release out of that perinuclear space.

These are radically different from roles traditionally associated with RNA helicases, which should open up new research avenues and innovative means to control viral replication.

## Methods

**Cells and viruses**. Hela (female human cervix; ATCC CCL-2), Vero (female African green monkey kidney; ATCC CCL-81), and HEK293 Flp-In (female human kidney; Jason Young, McGill University) cells were grown in DMEM (Dulbecco's Modified Eagle Medium) supplemented with 2 mM L- glutamine in 5% CO$_2$ and 5% bovine growth serum (BGS). The HSV-1 parental wild-type viral strains 17 + , F and KOS (provided by Beate Sodeik, Hannover Medical School) and a hemagglutinin (HA)-tagged U$_L$31 virus named vJB90 (F strain; obtained from Joel Baines;[54]) were grown and titrated on Vero cells. HSV-1 mutants and complementing cell lines were kind gifts from Prashant Desai (K5ΔZ; KOS strain; VP5-null virus[84]), Fred Homa (vFH421; KOS strain lacking the first 50 amino acids of U$_L$25 and only producing A- and B-nuclear capsids[34,82,83]), Joel Baines (v3161; U$_L$31 null virus[92]) and Richard Roller (vRR1072; F strain U$_L$34 null virus and vRR1202; Us3 null virus[58]). All viral mutants were amplified on complementing cell lines and titrated on Vero cells and compared to their respective wild-type viral strain. All cell lines were routinely tested and negative for mycoplasma.

**Antibodies**. Several laboratories generously provided some of the antibodies used in this study. These include anti-human DDX3X rabbit R648 polyclonal serum (1:100 in IF; 1:10,000 in WB and 1:300 for IP) from Arvind Patel[77] and anti-pU$_L$34 chicken polyclonal serum from Richard Roller (1:500 in IF)[43]. Joel Baines provided the pU$_L$31 rabbit polyclonal (1:1000 in WB) and anti-VP22 chicken polyclonal (1:10,000 in WB) sera[43,94]. Gill Elliott supplied the anti-VP13/14 rabbit polyclonal serum (1:5000 in WB)[95], Helena Browne the anti-VP16 mouse monoclonal antibody (1:1000 in WB)[96], Bernard Roizman the anti-pUs3 rabbit polyclonal serum (Bernard Roizman; 1:4000 in WB)[97] and Beate Sodeik and Ari Helenius both gave us some rabbit polyclonal serum against HSV-1 capsids (1:1000 in WB)[98]. Bruce Banfield provided us with pUs3 rat polyclonal serum that reacts against both HSV-1 and HSV-2 (1:500 in WB)[99]. Commercially available primary antibodies were also used: Anti-VP5 mouse monoclonal (EastCoast Bio; 1:1000 in WB), LBR mouse monoclonal antibody (Abcam Cat#ab232731; 1:50 in IF), GAPDH mouse monoclonal antibody (Millipore Sigma; 1:10,000 in WB), γ-Tubulin mouse monoclonal antibody (Millipore Sigma; 1:5000 in WB), HA-tag mouse monoclonal antibody (Santa Cruz; 1:100 in IF and 1:1000 in WB), mouse monoclonal ICP4 (Abcam; 1:200 in IF and 1:1000 in WB) and ICP0 (Abcam; 1:500 in WB). HRPO-Conjugated secondary antibodies were used in the concentration of 1:1000 and purchased from Bethyl Laboratories (Goat anti-Rabbit), Jackson ImmunoResearch (Goat anti-Mouse) or Cedarlane (Donkey anti-Chicken). For immunofluorescence, Goat anti-Chicken Alexa Fluor 488, Goat anti-Rabbit Alexa Fluor 568, Chicken anti-Mouse Alexa Fluor 488 and Donkey anti-Mouse Alexa Fluor 647 were purchased from Molecular Probes and used at a 1:1000 dilution. For STED microscopy, Goat-Anti-Rabbit-IgG-Atto 647 N (Sigma-Aldrich) and Goat anti-Mouse IgG (H + L) cross-adsorbed secondary antibody, Alexa Fluor 594 (Thermo Fisher Scientific) were used at the 1:100 dilution.

**Generation of DDX3X stable cell line**. HEK293 Flp-In cells, pOG44 and pcDNA5/FRT plasmids were generously provided by Dr. Jason Young (McGill University) and used to generate cells stably expressing GS-DDX3X (DDX3X fused to protein G) according to Thermo Fisher's guidelines. Briefly, GS-DDX3X was cut out from siRNA resistant pGS-DDX3X[25] and cloned into pcDNA5/FRT. HEK293 Flp-In cells were then co-transfected with pcDNA5/FRT- GS-DDX3X and pOG44 (1:10 ratio). Stable cells were selected using 200 μg/mL hygromycin B for about 4 weeks. This newly created cell line is available upon request.

**Isolation and proteomics of DDX3X viral binding partners**. HEK293 Flp-In GS-DDX3X expressing cells were grown on 150 mm dishes, mock-treated or infected with HSV-1 wild-type strain 17+ at a multiplicity of infection (MOI) of 5 for 4 h or 12 h. Cells were next lysed with 3 mL of lysis buffer (50 mM Tris-HCl pH 8.0, 150 mM NaCl, 1% NP-40, 10% glycerol) for 1 h at 4ºC. Lysates containing the tagged-DDX3X were enriched for GS-DDX3X using IgG Sepharose 6 Fast Flow affinity resins (GE Healthcare). Briefly, 3 mL of total lysate was incubated with 100 μL of IgG Sepharose overnight at 4ºC. IgG Sepharose was separated from the unbound material by centrifugation at 800 × g for 2 min and then washed three times in lysis buffer. Elution from the IgG Sepharose was performed by boiling (95 ºC) the beads in 100 μL of Laemmli sample buffer (50 mM Tris-HCl, pH 6.8, 2% SDS, 0.1% bromophenol blue, 10% glycerol and 2% β-mercaptoethanol) for 5 min. Eluates were separated on a 5–15% gradient SDS-PAGE gel and the bands revealed through colloidal Coomassie staining. The staining protocol was inspired by the Coomassie staining provided by the SERVA company with few modifications. For instance, gels were fixed (10% acetic acid and 40% ethanol in Milli-Q water) overnight at room temperature. Solution I (0.2% Coomassie Brilliant Blue G-250 (BioRad) in 90 % methanol) and solution II (20% acetic acid in Milli-Q water) were mixed in a 1:1 ratio and used to stain the gels for 20 min at room temperature. The gels were then rapidly destained for 5 min in the fixing buffer.

These gels were destained with a second solution (10% glacial acetic acid and 20% Methanol in Milli-Q water) until the desired contrast was achieved. Finally, the gels were washed twice in Milli-Q water. To prepare the samples for proteomics, the gels were cut into three sections: above the IgG heavy chain, between the IgG heavy and light chain and below the IgG light chain. These sections were conserved in Eppendorf tubes at −80 °C until their analysis.

Gel sections were shrunk in 50% acetonitrile (ACN), reconstituted in 50 mM ammonium bicarbonate with 10 mM TCEP (Tris(2-carboxyethyl) phosphine hydrochloride; Thermo Fisher Scientific, San Jose, CA) and vortexed for 1 h at 37 °C. Alkylation was performed with 55 mM chloroacetamide (Sigma-Aldrich) for an hour at 37 °C with vortexing. Samples were next digested for 8 h at 37 °C with 1 μg of trypsin and peptides subsequently extracted using 90% ACN. The extracted peptides were dried down and solubilized in 5% ACN-0.2% formic acid (FA). Digested peptides were loaded on a home-made C4 precolumn (Optimize Technologies) and separated on a home-made reversed-phase column (150 μm i.d. by 150 mm) with a 56 min gradient from 10 to 30% ACN-0.2% FA and a 600 nl/min flow rate on an EasynLC 1000 (Thermo Fisher Scientific, San Jose, CA) connected to an LTQ-Orbitrap Fusion (Thermo Fisher Scientific, San Jose, CA). All the full MS spectrums were acquired at a resolution of 60,000 and followed by tandem-MS (MS-MS) spectra acquisition on the most abundant multiply charged precursor ions for a maximum of 3 s. A collision-induced dissociation (CID) at a collision energy of 30% was performed for tandem-MS experiments. PEAKS X (Bioinformatics solutions, Waterloo, ON) and a concatenated UniProt human and herpesvirus database were used to process the data. Mass tolerances on precursor of 10 ppm and fragment ions of 0.01 Da were chosen. Variable selected post-translational modifications were carbamidomethyl (C), oxidation (M), deamidation (NQ), and phosphorylation (STY). The obtained data were analyzed with Scaffold 4.3.0 (protein threshold, 95%, with at least 2 peptides identified and a false-discovery rate of 0.1% for peptides). The data are available via ProteomeXchange with identifier PXD03940.

**Co-immunoprecipitations**. To confirm the physical interaction between DDX3X and pUL31, HeLa cells were grown on 100 mm dishes, mock-treated or infected with HA-tagged $U_L31$ virus (vJB90) or wild-type strain 17+ at an MOI of 5 for 12 h. Cells were lysed in 1.5 mL of RIPA buffer (50 mM Tris, pH 7.5, 150 mM NaCl, 1 mM EDTA, 1% Triton X-100)[100] for 30 min at 4 °C. The lysates were sonicated with 10 ×1 s pulses at intensity of 8 in a Microcup-horn sonicator at 4 °C. The sonicated lysates were precleared using mouse pre-immune serum and 50 μl of protein A agarose. To immunoprecipitate HA-pUL31, fresh 50 μl aliquots of Pierce anti-HA agarose (Thermo Fisher) were blocked using 1% BSA in RIPA buffer overnight at 4 °C. Blocked beads and the precleared lysates were incubated together at 4 °C for 3 h. Beads were pelleted by centrifugation at 12,000 × g for 10 s and then washed three times in RIPA buffer. Bead-bound proteins were eluted from the beads by boiling in 50 μl of sample buffer for 5 min. Alternatively, to immuno-precipitate DDX3X, the lysates were precleared using a rabbit pre-immune serum (kindly provided by Susanne Bailer) and 100 μL of protein A agarose (Millipore Sigma) for 1 h at 4 °C. In parallel, 100 μL of protein A agarose were incubated overnight with anti-human DDX3 rabbit R648 polyclonal serum at 4 °C. The precleared lysates were added to anti-DDX3 R648 conjugated A agarose beads and for 3 h at 4 °C. As a control, one of the infected precleared lysates was incubated on the beads without the primary antibody (no first antibody (NFA)). Beads were washed with the lysis buffer three times. To elute the bead-bound proteins, beads were boiled (95 °C) for 5 min in 100 μL of Laemmli sample buffer. All samples were analyzed through Western blotting as detailed below.

In order to distinguish the exact cellular site where DDX3X and pUs3 interact, we fractionated our cells into cytoplasm, nuclear matrix and nuclear membranes prior to co-IP. For this, HeLa cells were seeded in 150 mm dishes 24 h before infection and then infected with 17+ at an MOI of 5 or mock treated. At 12 hpi, cells were lysed by resuspending in 3 mL of mild lysis buffer (10 mM Tris, pH 7.4, 150 mM NaCl, 2 mM MgCl₂ 1 mM EDTA, 1 mM DDT and 0.5 % Igepal) and incubated on ice for 10 min, followed by vortexing for 30 s. Lysates were then centrifuged at 500 × g for 5 min to separate the cytoplasm from nuclei. Once the supernatant harvested, the nuclear pellets were resuspended in a lysis buffer without any detergent (50 mM Tris, pH 7.5, 150 mM NaCl, 1 mM EDTA) and three cycles of freeze-thaw performed to break the nuclei and release the nuclear matrix. This mix was then centrifuged at 2500 × g for 10 min to separate the nuclear membranes from the matrix. Finally, the pellet (nuclear membranes) was resuspended in the RIPA buffer mentioned earlier and incubated on ice for 30 min. This was followed by three passages through a 27 ½ G needle to shear the cellular genomes. Fractions were tested for purity by Western blotting against GAPDH and LBR. Infected fractions were then used to perform IP against DDX3X using the anti-DDX3X R648 as explained above. Lastly, total lysates and eluates were examined by WB for the presence of pUs3 and enrichment of DDX3X.

**Mapping of pUL31 and recombinant GST-tagged DDX3X fragments**. DDX3X is a 632 amino acid (aa) protein. pGEX-DDX3X bacterial constructs expressing various DDX3X fragments were generously provided by Dr. Arvind Patel. These included pGEX-4T-DDX3X aa1-141, pGEX-4T-DDX3X aa1-208, pGEX-4T-DDX3X aa1-282, pGEX-4T-DDX3X aa1-472, pGEX-6P-DDX3X aa472-662, pGEX-6P-DDX3X aa498-662 and pGEX-6P-DDX3X aa623-662.

BL21-CodonPlus (DE3) -RIPL bacteria (Agilent Technologies) expressing the GST-tagged DDX3X fragments mentioned earlier, and the corresponding empty

pGEX-6P-1 vector (provided by Dr. Marino Zerial) were grown in 40 mL of Luria-Bertani (LB) at 37 °C until the OD₆₀₀ reached 0.4. The expression of the fragments was induced using 1 mM of isopropyl β- d-1-thiogalactopyranoside (IPTG) at 37 °C for 4 h. The samples were centrifuged at 4000 x g for 20 min at 4 °C. Bacterial pellets were resuspended in the bacterial lysis buffer (20 mM Tris-HCl pH 7.5, 500 mM NaCl, 1 mM EDTA, 1 mM EDTA, protease inhibitors and 1 mg/mL of lysozyme (Sigma)) at 4 °C for 1 h. Samples were subsequently sonicated on a Fisherbrand sonicator with a probe (3x pulses of 5 s ON and 5 s OFF at intensity 8 and then 7x pulses of 5 s ON and 5 s OFF at intensity 7). The lysates were cleared at 8500 x g for 20 min at 4 °C and incubated at 4 °C for 3 h with 100 μL of Glutathione Sepharose 4B (Cytiva). Meanwhile, HeLa cells were seeded on 150 mm dishes (about 7 × 10⁶ cells per GST-tagged construct), infected with the 31-HA virus at an MOI of 5 for 24 h, then lysed in RIPA buffer. Cellular lysates were precleared on GST (bound to the Glutathione Sepharose) for 3 h. The precleared cellular lysates were then divided equally between the different Glutathione Sepharose bound GST-tagged constructs. Samples were incubated for 3 h at 4 °C. Finally, the Sepharose was washed 3 times with lysis buffer and pelleted at 500 x g. To release the bait, the Sepharose was boiled in 100 μL Laemmli sample buffer. All samples were analyzed by Western blotting as indicated below.

**Western blotting**. To prepare the samples for Western blotting, we performed BCA assays (Pierce BCA assay kit; Thermo Fisher Scientific) or Bradford Bio-Rad protein assays to determine protein concentrations. In general, 25 μg of each sample was loaded on 5–15% SDS-PAGE gradient gels. To visualize total proteins, 0.5% of 2,2,2-Trichloroethanol (TCE; Sigma) incorporated in the gels was photo-activated on a ChemiDoc (BioRad) using the stain-free gel application (590/110 filter and 45 s of activation). Proteins were next transferred to polyvinylidene difluoride (PVDF) membranes and then blocked in milk blocking buffer (5% non-fat dry milk in 13.7 mM NaCl, 0.27 mM KCl, 0.2 mM KH₂PO₄, 1 mM Na2HPO₄, 0.1% Tween 20) at room temperature for 1 h. Subsequently, the blocked PVDFs were reacted with the appropriate primary antibody diluted in BSA blocking buffer (5% bovine serum albumin in 13.7 mM NaCl, 0.27 mM KCl, 0.2 mM KH₂PO₄, 1 mM Na2HPO₄, 0.1% Tween 20) overnight at 4ºC. Finally, the PVDFs were reacted with the HRPO-conjugated secondary antibodies diluted in 5% milk blocking buffer and revealed using Clarity Western ECL substrate (Bio-Rad) on a Syngene gel documentation (gel doc) system. Where indicated, images were quantified using Image J software (NIH).

**Standard confocal fluorescence microscopy**. Cells were seeded on round glass coverslips (Fisherbrand) and treated according to each individual experimental setup. They were then fixed in 4% paraformaldehyde (PFA) in phosphate-buffered saline (PBS) for 15 min at room temperature. All the reagents mentioned below were prepared based on the immunofluorescence (IF) protocol provided by Cell Signaling Technologies. Fixed samples were blocked for 1 h at room temperature in IF blocking buffer (1X PBS, 5% normal serum and 0.3% Triton X-100). Afterwards, cells were reacted overnight at 4 °C with the appropriate primary antibody diluted in the IF antibody dilution buffer (1X PBS, 1%, BSA and 0.3% Triton™ X-100). Cells were washed three times in 1X PBS and then incubated with the appropriate secondary antibody in IF antibody dilution buffer for 1 h at room temperature. Cells were finally washed three times using 1X PBS and were mounted on glass slides in Dako containing 0.1 μg/mL Hoechst 33342 (Sigma-Aldrich). Imaging was performed using a LSM800 confocal microscope (Zeiss) or a Leica TCS SP8-DLS (TCS SP8 laser scanning confocal microscope with DLS light sheet module).

**Stimulated emission depletion (STED) microscopy**. HeLa cells were grown on precision cover glasses with #1.5H thickness (ThorLabs). After various treatment according to the specific experimental protocol, cells were fixed in 4% PFA for 15 min at room temperature. The buffers used for blocking and antibody dilution are the same as the ones mentioned for standard confocal fluorescence microscopy, but primary anti-bodies were used in two-fold higher concentrations than for conventional confocal fluorescence microscopy to ensure the efficient detection of the targets. All samples were incubated with primary antibodies overnight at 4 °C. STED compatible secondary antibodies were incubated for 2 h at room temperature. Coverslips were then mounted on microscopic slides containing 0.1 μg/mL Hoechst 33342 in ProLong Glass antifade (Thermo Fisher Scientific). Samples were dried at room temperature overnight. Imaging was done on a laser scanning confocal microscope (Leica TCS SP8) equipped with white light laser for excitation wavelength between 470 and 670 nm and two depletion lasers at 592 and 775 nm. These images were further processed by LIGHTNING mode on the LAS X software for increased resolution.

**Analysis of DDX3X and LBR colocalisation with Imaris software**. HeLa cells seeded on coverslips were mock-treated or infected with wild-type strain 17+ at an MOI of 5 for 9 h. DDX3X (Goat-Anti-Rabbit-Atto 647 N) and LBR (Goat anti-Mouse-Alexa Fluor 594) were labelled as above (STED microscopy) and the nuclei stained with 0.1 μg/mL Hoechst 33342. Z-stack images were taken on a Leica TCS SP8-STED using a HC PL APO 100x/1.40 CS2 objective with laser intensities adjusted to avoid signal saturation. For each condition, twelve random cells were imaged. All the following analyses were then performed on the Imaris software version 9.8.0. First, the surface wizard was chosen to process the LBR signal. First,

in the thresholding option, the background subtraction was chosen so that objects <0.1 μm would be eliminated. Next, any signal below 50 voxels was eliminated from the surface. Lastly, the intensity mean of the LBR signal was set to 500, based on the visual observations to achieve the cleanest surface. All these three parameters were kept the same for all the images throughout the analyses. A binary mask was created on the LBR surface. The only processing step for the DDX3X channel was a background subtraction to eliminate objects <0.1 μm. To avoid any inaccuracy in the colocalisation measurement, only few middle z-stacks (based on the number of stacks which can vary between images) where a clear nuclear rim shape was observed were chosen. Finally. using the Coloc wizard in Imaris a colocalisation channel was built. In the "coloc" estimated statistics, the number represented in the % of ROI colocalised was chosen to demonstrate the percentage of DDX3X signal found on the masked signal of LBR surface.

**Analysis of VP5 phenotypes with Imaris software**. HeLa cells seeded on coverslips were transfected with 25 nM of siDDX3X (Dharmacon cat # D-006874-01-0002) or NC1 (IDT) for 48 h as described previously[25] and then mock-treated or infected with wild-type strain 17$^+$ at an MOI of 5 for 9 h. VP5 and DDX3X were labelled as above (standard confocal microscopy) and the nuclei stained with 0.1 μg/mL Hoechst 33342. Z-stack images were taken on Leica TCS SP8-DLS using a HC PL APO CS2 63x/1.40 oil objective and 12-bit depth with laser intensities adjusted to avoid signal saturation. For each condition, 20 cells from six to eight random images were chosen. All the following analyses were then performed on the Imaris software version 9.7.2. First, the surface wizard was chosen to process the VP5 signal. Surface was smoothed using the Gaussian filter and background subtraction options. Threshold was adjusted so that a small halo can be observed around the surface and touching objects were separated by adjusting the seed point diameter. Finally, any signal below five voxels was eliminated from the surface. To define the nuclei, we used the Gaussian filter with all other parameters left untouched. Cell boundaries were delimited with the click-draw option. The DDX3X signal was used to draw a border around each cell. Object to object statistics were enabled to analyze the VP5 signal (signal intensity sum) and position (μg) in relation to the nuclear envelope. The VP5 % intensity sum is the ratio of each object to the total VP5 signal in the cell. These sums were then averaged over 20 cells in each of three independent experiments.

**Extracellular virions and nuclear capsids purification**. Mature HSV-1 particles released in the extracellular medium and nuclear capsids were purified from HeLa cells as described previously[101]. Briefly, to produce the extracellular virions, HeLa cells were infected with wild-type strain 17$^+$ at an MOI of 5 for 24 hpi. Extracellular medium was filtered on 0.45 μm filter and then concentrated at 60,000 x g. For the nuclear capsids, intact nuclei were isolated from these infected cells and the nuclear capsids released from the nuclei with three cycles of freeze/thaw[29,31,86]. Total nuclear capsids were enriched on a 35% sucrose cushion by centrifugation at 100,000 x g.

**Electron microscopy**. HeLa cells seeded on 6-well plates at the concentration of 11,000 cells per well 24 h prior to treatment were treated with 25 nM of siDDX3X or the NC1 non-targeting control for 48 h. Subsequently, transfected HeLa cells were either mock treated or infected with wild-type strain 17$^+$ at an MOI of 5 for 9 or 12 h. Cells were then washed twice in 0.1 M sodium cacodylate buffer (pH 7.2) and then fixed for 1 h at 4 °C (2.5% glutaraldehyde, 2% paraformaldehyde, 0.1 M cacodylate buffer, pH 7.2). Fixed samples were washed twice in 0.1 M cacodylate buffer, pH 7.2, scraped and pelleted by centrifugation at 3300 x g. The pellets were resuspended in post-fixation buffer (1% osmium tetroxide, 0.1 M cacodylate buffer) for 1 h at 4 °C. Afterwards, a gradual dehydration of cells was performed using ethanol at 30%, 50%, 70%, 95%, and 100%. Finally, using propylene oxide, cells were permeabilized and then embedded in Epon (Epon 812; dodecenyl succinic anhydride (DDSA), nadic methyl anhydride (NMA) plus tri (dimethyl amino methyl) phenol (DMP-30)). Thin sections were prepared with a Leica (MZ6) Ultracut UCT ultramicrotome (80–90 nm thickness) and imaging performed on a Phillips 300 or Philips Tecnai 12 transmission electron microscope.

**Tegument content of extracellular vrions**. HeLa cells were seeded in six 245 mm square dishes 24 h prior to infection. Once the cells reached 90% confluence, they were infected with wild-type strain 17$^+$ virus at an MOI of 5 for 18 h. The infected supernatants were collected and pulled, filtered through a 0.45-μm cellulose acetate filter (Costar) and then centrifuged at 500 x g for 5 min at 4 °C. The extracellular virions were pelleted from the supernatant by centrifugation at 20,000 x g for 1 h at 4 °C. The pellet was washed in MNT (30 mM morpholineethanesulfonic acid, 100 mM NaCl, and 20 mM Tris-HCl, pH 7.4) by another round of at 20,000 x g for 1 h at 4 °C. The washed pellet was resuspended in MNT overnight at 4 °C to maximize viral recovery. About 25 μg of extracellular virions, mock and infected total lysates (HeLa cells mock treated or infected wild-type strain 17$^+$ virus at an MOI of 5 for 18 h and then lysed in RIPA buffer) were analyzed by Western blotting using a bank of viral antibodies.

**Statistics and reproducibility**. Where applicable, protein abundance and fluorescence intensities were normalized to the values calculated for the controls as described in each figure legend and analyzed with bilateral Student's T-tests (two-tailed distribution and homoscedastic) on Excel (Microsoft Office 365). The significance threshold was set to P-value of <0.05. All experiments were independently performed three times prior to statistical analysis.

**Reporting summary**. Further information on research design is available in the Nature Portfolio Reporting Summary linked to this article.

## Data availability

Raw data are provided as Supplementary Data 1, while original uncropped blots can be found in Fig. S9. The mass spectrometry proteomics data have been deposited to the ProteomeXchange Consortium via the PRIDE[102] partner repository with the dataset identifier PXD039403 and 10.6019/PXD039403 (https://www.ebi.ac.uk/pride/archive/projects/PXD039403).

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

## Acknowledgements

We are indebted to the numerous people that provided viruses, antibodies and plasmids (Drs. Joel Baines, Bruce Banfield, Helena Browne, Prashant Desai, Gill Elliott, Ari Helenius, Fred Homa, Arvind Patel, Bernard Roizman, Richard Roller, Beate Sodeik and Jason Young). We also wish to thank Diane Gingras (University of Montreal) and Elke Küster-Schöck (imaging platform at the Sainte-Justine Research Center) for their invaluable help with electron and fluorescence microscopy respectively. Finally, we would like to thank Matthew Gastinger (bitplane) for assistance in fluorescent image analysis and Michael Way and Arvind Patel for their critical reading of the manuscript and all the help the latter provided with his DDX3X expertise. This research was possible thanks to funds from the Canadian Institutes of Health Research (CIHR; MOP 82921 & PJT-178115), the Sainte-Justine Foundation and past equipment funding from the Canadian Foundation for Innovation (CFI) and the Natural Sciences and Engineering Research Council of Canada (NSERC; RGPIN-2016-04277). The Institute for Research in Immunology and Cancer (IRIC) receives infrastructure support from IRICoR, the Canadian Foundation for Innovation, and the Fonds de Recherche du Québec–Santé (FRQS). IRIC proteomics facility is a Genomics Technology platform funded in part by Genome Canada and Genome Québec. B.K. is the recipient of a PhD award from the Sainte-Justine Foundation.

## Author contributions

Conceptualization, R.L. and B.K.; methodology, B.K., E.B. P.T., and R.L.; formal analysis, B.K., E.B., P.T. and R.L.; investigation, B.K. and E.B.; resources, R.L., P.T.; writing—original draft, R.L. and B.K.; writing—review and editing, B.K., E.B. P.T., and R.L.; visualization, B.K.; supervision, R.L. and P.T.; project administration, R.L., E.B., and P.T.; funding acquisition, R.L. and P.T.

## Competing interests

The authors declare no competing interests.

## Additional information

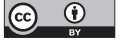

