## [Peer Review File · Communications Biology]

Reviewers' comments:

Reviewer #1 (Remarks to the Author):

The paper by Khadivjam et al. titled "RNA helicase DDX3X modulates Herpes Simplex Virus 1 nuclear egress" describes the mechanistic role of DDX3X to facilitate HSV1 propagation. Data is provided to indicate that DDX3X physically interacts with virally encoded nuclear egress complex as well as promotes viral nuclear release via binding to mature herpes kinase, pUs3. These interactions between DDX3X and HSV-1 are essential to complete the viral life cycle. Overall, the scientific rigor is high with the appropriate controls. Some of the concerns that need to be addressed are:

- 1) Does knockdown of DDX3 by siRNA alter the proliferation rate of the host cells? Cell cycle analysis of these transgenic cells should be provided.
- 2) Should demonstrate the number of virions released in the extracellular environment, following DDX3X knockdown, is decreased compared to the control siRNA treated cells.
- 3) Confirmatory experiments with DDX3X inhibitors should be carried for potential translation utility.

Reviewer #2 (Remarks to the Author):

Herpes simplex virus (HSV) is involved in various mucocutaneous and neurological diseases. Although anti-viral drugs are available for herpesviruses, but these drugs cannot completely eradicate the virus from the host. Thus, fundamental research to uncover how HSV replicates within the cells is essential to establish anti-viral strategies in the future. Nuclear egress is the nucleo-cytoplasmic transport of newly synthesized capsids within the nucleus, which is unique in cell biology. In this manuscript, the authors analyzed the role of the mammalian RNA helicase DDX3X for HSV nuclear egress. The authors showed that depletion of DDX3X exhibited a defect in nuclear egress. DDX3X interacted with viral protein UL31, which is the component of nuclear egress complex (NEC). Viral protein kinase Us3 has been considered to promote nuclear egress by disassembling NEC lattices. The authors showed that DDX3X is important for incorporating Us3 into the perinuclear vesicles produced by NEC in infected cells. This manuscript is well-written and will be of interest to the broader readership of this journal. However, a few comments are listed below to improve this manuscript before publication.

(i) Fig. S1 shows localization and accumulation of the tagged DDX3X only in uninfected cells. The authors should show the localization of the tagged DDX3X in infected cells.

(ii) In Fig. S5, how can you determine the percentage of nuclear membrane-localized DDX3X? UL34 seems colocalized with DDX3X in UL31 delta virus-infected cells. If so, DDX3X rather interact with UL34 directly.

(iii) In Fig. 7 and Fig. S5, the localization of UL34 in UL31 deletion is not convincing as UL34 is a membrane protein that localizes at ER primary.

(iv) Fig. 10, can the authors purify and do the same experiment using perinuclear virions? Immunoblot of UL31, UL34, and US3 should be shown in DDX3X depletion.

(v) Throughout the manuscript, the authors analyzed the role of DDX3X using a knockdown experiment. As mentioned in the introduction, DDX3X is the RNA helicase, and it should be better to show the importance of this enzymatic activity and the role of RNA during nuclear egress. A similar system was also observed during a large RNP transport in uninfected *Drosophila* cells. Thus, it would be much more interesting for broader readers if the authors showed the role of RNA during nuclear

egress in viral capsid.

Minor comments.

(i) In Line 34, the word "highjack" is unsuitable. The data in this manuscript shows the importance of DDX3X for viral replication but does not show that viruses utilize the function of DDX3X.

(ii) Lines 168-169, please indicate UL47 also precipitated as it is essential for nuclear egress of HSV-1 (Liu et al., JVI, 2014). Also, please indicate that UL34 was not detected as UL31 and UL34 make a stable complex in infected cells.

Reviewer #3 (Remarks to the Author):

Summary

Nuclear egress is an unusual kind of nuclear export that herpesviruses use to transport their capsids from the nucleus into the cytoplasm during infection. Although the main viral proteins that mediate this process have been identified, less is known about the host participants. In this manuscript, Khadivjam et al report that the cellular DEAD-box RNA helicase DDX3X appears to contribute to the nuclear egress of the prototypical herpes simplex virus 1 (HSV-1). This is an unexpected finding because DEAD-box RNA helicases are ATP-dependent enzymes that unwind misfolded RNA to help it refold. But how such activity is involved in the nuclear egress process is not obvious. Nonetheless, the authors present evidence that links DDX3X to HSV-1 nuclear egress. They show that during infection, DDX3X relocalizes from the cytoplasm to the perinuclear region and that in the absence of DDX3X, the nuclear egress of capsids is reduced. They also show that DDX3X interacts with the viral protein UL31, which is a component of the viral nuclear egress complex (NEC) that mediates the nuclear budding stage of the egress. This interaction, be it direct or indirect, could explain why relocation of DDX3X to the perinuclear region requires UL31.

Less clear is the precise role of the DDX3X in nuclear egress. Authors propose that DDX3X facilitates de-envelopment of the perinuclear enveloped virions (PEVs, which are formed as the result of NEC-mediated nuclear budding of the capsids) by promoting the incorporation of the HSV-1 kinase US3 into the PEVs. But this contention is not sufficiently supported by the presented data. The authors also observe large foci on the nuclear periphery that are enriched in the major capsid protein and do not form in the absence of DDX3X, but the nature of these foci remains uncertain. As the result, while the discovery of the potential involvement of a host RNA helicase in HSV-1 nuclear egress is interesting, the proposed mechanism is confusing and is not sufficiently justified. The paper is mostly well written, and many experiments, especially, the confocal images showing co-localization, are convincing and are done with proper controls. But there are some concerns regarding the experiments. Some important controls are lacking. For example, the authors do not confirm that the DDX3X siRNA treatment, indeed, reduces protein levels. Additionally, several Western Blot images look odd, potentially, due to image manipulation.

Major criticisms

1. Line 78-82: The authors contradict themselves here. If NEC phosphorylation were a negative regulator of nuclear budding, then in the absence of phosphorylation by the US3 kinase, one would expect to see increased rather than decreased budding. Indeed, the accumulation of PEVs, which are the products of budding, is consistent with increased budding. The accumulation of PEVs in this and other viral mutants or host protein knockdowns could be simply due to their overproduction rather than an additional defect in de-envelopment. This is an important point that the authors should consider in interpreting their data and proposing models.

2. Line 132-133, 232-233, 312, and elsewhere: Authors need to confirm the DDX3X knockdown experimentally.

3. Line 203: The conclusion that the UL31 deletion mutant failed to recruit DDX3X to the nuclear envelope is not justified. Fig. S5B shows a 3-fold reduction in colocalization. Moreover, currently, it is challenging to assess colocalization (or lack thereof) in Fig S5. Colocalization should be quantified as in Fig. 2.

4. Lines 252-254: The conclusion that the large VP5-containing foci are C-capsid-dependent is not justified. Fig. 7 clearly shows large foci in the Δ 1-50 UL25 mutant even though the number of cells containing such foci is reduced.

5. Line 259-260 and Fig. 8A. The nuclear localization of UL34 in Δ VP5 and Δ 1-50 UL25 mutants is disconcerting. Why is UL34 inside the nucleus? Is it lacking its transmembrane anchor? Even in transfected cells where no other viral proteins are present, UL34 localizes to the nuclear envelope in the presence of UL31.

6. Authors report that DDX3X colocalizes with VP5 in large foci at the nuclear periphery that are seen in 20% of the infected cells (Fig. 5). These foci disappear when DDX3X is knocked down (Fig. 6). What are these large foci? They are unlikely to be nuclear envelope herniations containing PEVs because these structures increase in the absence of DDX3X (Fig. 3). They can form even in the absence of C-capsids albeit in a smaller number of cells (Fig. 7). In a contradictory manner, the authors then propose that the nuclear envelope herniations containing PEVs (Fig. 9B) correspond to the large foci observed by confocal microscopy. What are these foci? Are they the herniations or not? At any rate, it is important to note that the herniations are very rare and quite small in cells infected with the WT HSV-1.

7. Line 296-297 and 323-325: The conclusion that DDX3X promotes the US3 incorporation into the PEVs or that this facilitates the de-envelopment is not sufficiently supported by the presented data. Co-immunoprecipitation of UL31 and US3 from nuclear membranes (Fig. 11) only tells us that they may interact there, be it directly or indirectly. It does not confirm that DDX3X facilitates the incorporation of US3 into PEVs. Likewise, de-envelopment was not assayed here. PEV accumulation can be due to their overproduction.

8. Western Blot panels in Fig. 4A, 10A, 11A, and 11B look odd. Was this image manipulated in any way? If so, this needs to be explained.

Minor criticisms

9. Line 45-52: What is known about the DDX3X mechanism in the host and in viruses listed here? Can its RNA helicase activity account for its involvement in viral replication? If not, Does DDX3X have any activities other than RNA unwinding? It is important to describe these here because non-helicase activities if any, could provide clues into its potential role in HSV nuclear egress.

10. Line 36 and throughout: Eliminate "physical" in reference to protein interactions. Protein interactions can be direct or indirect. The term "physical" is nebulous.

11. Line 54-55: How does DDX3X influence HSV gene expression? Please, clarify.

12. Line 61-63: All 4 types of capsids are referred to as viral intermediates. Does this refer to an intermediate in the virion assembly? Please, clarify.

13. Line 75: What is meant here by "the smooth distribution of the NEC components around the nucleus"? If this is in reference to the uneven distribution of the immunofluorescent signal of the NEC components in the absence of US3, then this reflects the accumulation of the NEC-coated PEVs in perinuclear herniations rather than the relocalization of UL31 and UL34 along the nuclear envelope.

Please, clarify.

14. Line 82-83: Several proteins, e.g., SLC35E1, are missing from the list of host proteins that have been implicated in herpesvirus nuclear egress. ESCRT is not a protein but a class of protein complexes, ESCRT-0, -I, -II, and -III. Which component is referred to here?
15. Line 83-84. Unnatural sentence. Please, rephrase.
16. Line 89: replace "let" with "led".
17. Line 94, 128, and elsewhere: replace "repositioning" with "relocalization".
18. Line 95 and elsewhere: none of the methods used in this work establish that the DDX3X and UL31 interact directly, so this should be stated.
19. Line 100: replace virions with PEVs or an equivalent term, to distinguish these intermediates from mature, infectious virions.
20. Line 101-102: Is DDX3X present in PEVs?
21. Line 110: Replace "nucleus" with "nuclear envelope".
22. Line 127, 140-141, 296-297, 301-302, and elsewhere: The conclusion that DDX3X modulates PEV de-envelopment is not supported by the data. De-envelopment was not assayed here. PEV accumulation can be due to their overproduction (see comment 1). The presented data only show that in the absence of DDX3X, just as in the absence of US3, PEVs accumulate.
23. Line 134: Surely, extracellular virions are not viral assembly intermediates.
24. Line 150-164: Here, it would be helpful to briefly explain that this was a pulldown experiment.
25. Line 172-174. Was this a pull down?
26. Line 183: What is meant by "partial" interaction here? How was this assessed? The colocalization shown in Fig. S3 should be quantified as in Fig. 2.
27. Lines 185-197: The binding experiment setup needs to be explained upfront.
28. Line 209: Explain what is meant by the "smooth positioning" here.
29. Line 215: This section is very long and hard to follow. Consider breaking it up into several sections to increase readability.
30. Line 222-235: Authors report that DDX3X colocalizes with VP5 in large foci at the nuclear periphery that are seen in 20% of the infected cells, but in Fig 5B, not all such foci appear to have a clear DDX3X signal. Colocalization should be quantified.
31. Line 235-242: What kind of capsid transport or aggregate transport is measured here (Fig. 6C)? Please, explain the rationale and the results better.
32. Lines 305-308: The rationale for assaying extracellular virions for tegument proteins as a readout for nuclear egress is puzzling. Tegument proteins are present in the cytoplasm and can be recruited there. Even if the amount of US3 in extracellular virions is somewhat lower in the absence of DDX3X (Fig. 10), this does not tell us anything about its incorporation into the PEVs.

33. There are too many figures. Consider combining Figs 1 and 2, 10 and 11, S1 and S2, and moving 6A, 6C, and 7A into SI.

34. Fig. 1: the size of the nucleus in the 9h time point appears larger. Is the scale the same for all timepoints?

35. Fig. 3: In panels AB, consider showing boxes in the zoomed-out panels to clarify where magnified images come from. In panel C, what do N1 and N2 refer to?

36. Fig 4: Was a reverse co-immunoprecipitation using HA-tagged pUL31 done? Additionally, the DDX3X bands in 4C do not match those in Fig S4.

37. Fig 5: does the DDX3X colocalize with gB within the large foci?

38. Fig 9: Two different WT stains KOV and 17+ were used in different experiments. The authors need to confirm that both strains behave similarly with regard to the phenotypes described here.

Reviewer #1 (Remarks to the Author):

The paper by Khadivjam et al. titled “ RNA helicase DDX3X modulates Herpes Simplex Virus 1 nuclear egress” describes the mechanistic role of DDX3X to facilitate HSV1 propagation. Data is provided to indicate that DDX3X physically interacts with virally encoded nuclear egress complex as well as promotes viral nuclear release via binding to mature herpes kinase, pUs3. These interactions between DDX3X and HSV-1 are essential to complete the viral life cycle. Overall, the scientific rigor is high with the appropriate controls. Some of the concerns that need to be addressed are:

Many thanks for pointing out the scientific rigor of our manuscript. This is appreciated.

1) Does knockdown of DDX3 by siRNA alter the proliferation rate of the host cells? Cell cycle analysis of these transgenic cells should be provided.

This is an important point. We have been using the same siDDX3X reagents since 2009 and addressed its impact on cells at that time. This information is already published and shows that the siDDX3X do not significantly impact cell viability (80% viability; Stegen, Plos One 2013). We again used those same siDDX3X reagents in our 2017 paper with no apparent effect on cell viability over the 3-day experiments (Khadivjam, J Virol 2017). It is thus clear that siRNA toxicity is not an issue. We clarified this point in the manuscript.

2) Should demonstrate the number of virions released in the extracellular environment, following DDX3X knockdown, is decreased compared to the control siRNA treated cells.

Thanks for pointing that omission. We indeed previously published this information in two separate papers (Stegen, Plos One 2013; Khadivjam, J Virol 2017). Depletion of DDX3X by RNA interference reduces extracellular viral yields by roughly 50 to 70%. This information was clarified in the manuscript.

3) Confirmatory experiments with DDX3X inhibitors should be carried for potential translation utility.

This is an exciting avenue that we are currently testing. Note that there are several dozens of such inhibitors, that we would need to compare to insure specificity. Given the large size of the current study (19 composite figures including the supplementary data), this plan this subject for a future publication with different authors (the first author has now graduated). So far, this looks promising as our data indicates that one of those inhibitors can reduce viral yields by up to 90%.

Reviewer #2 (Remarks to the Author):

Herpes simplex virus (HSV) is involved in various mucocutaneous and neurological diseases. Although anti-viral drugs are available for herpesviruses, but these drugs cannot completely eradicate the virus from the host. Thus, fundamental research to uncover how HSV replicates within the cells is essential to establish anti-viral strategies in the future. Nuclear egress is the nucleo-cytoplasmic transport of newly synthesized capsids within the nucleus, which is unique in cell biology. In this manuscript, the authors analyzed the role of the mammalian RNA helicase DDX3X for HSV nuclear egress. The authors showed that depletion of DDX3X exhibited a defect in nuclear egress. DDX3X interacted with viral protein UL31, which is the component of nuclear egress complex (NEC). Viral protein kinase Us3 has been considered to promote nuclear egress by disassembling NEC lattices. The authors showed that DDX3X is important for incorporating Us3 into the perinuclear vesicles produced by NEC in infected cells. This manuscript is well-written and will be of interest to the broader readership of this journal. However, a few comments are listed below to improve this manuscript before publication.

Thanks for pointing out that the manuscript is well-written and of interest to this journal.

(i) Fig. S1 shows localisation and accumulation of the tagged DDX3X only in uninfected cells. The authors should show the localisation of the tagged DDX3X in infected cells.

The intent of figure S1 is to show that the GS tag does not influence the localisation of DDX3X and that its expression is similar to that of the endogenous DDX3X, which are potential caveats of exogenously expressed genes. The data indeed show that both GS-DDX3X localisation and expression levels are normal. The same was previously noted in a different cell line, namely HeLa cells (Khadivjam, J Virol 2017). That study also showed that GS-DDX3X behaves as and can even functionally complement the endogenous DDX3X in infected cells. This already published information was clarified in the manuscript. Note that HEK293 Flp-In cells are difficult to analyze by fluorescence microscopy at the virus perturbs their morphology and especially shrink their cytoplasm.

(ii) In Fig. S5, how can you determine the percentage of nuclear membrane-localised DDX3X? UL34 seems colocalised with DDX3X in UL31 delta virus-infected cells. If so, DDX3X rather interact with UL34 directly.

As indicated on the Y-axis, we counted the number of cells that exhibited DDX3X nuclear staining. We now added that information in the legend.

As pointed by the reviewer, we cannot rule out an indirect binding of DDX3X to UL31 that may occur through UL34. The statistically significant reduction in DDX3X nuclear localisation observed in the absence of UL31, while UL34 remained nuclear bound, however favors some UL31 implication. The intriguing absence of pUL34 in our MS analysis, despite its usual binding to pUL31, further corroborates this view. In contrast,

that MS analysis identified pUL47, a protein that interacts with the NEC and mediates HSV-1 nuclear egress (Liu, J Virol 2014) and could bind DDX3X (also see minor ii below). It will be most interesting to resolve its potential role as well as the importance of NEC phosphorylation on DDX3X nuclear binding. We now discuss these scenarios in the discussion.

(iii) In Fig. 7 and Fig. S5, the localisation of UL34 in UL31 deletion is not convincing as UL34 is a membrane protein that localises at ER primary.

It's been known that the nuclear positioning of UL34 is both UL31 and US3 dependent. Fig S5 shows that UL34 is partly nuclear membrane bound but spreads over an ER-like compartment in the absence of UL31, suggesting some erroneous membrane targeting. This is in fact in full agreement with the original report by Richard Roller's lab on UL31 and UL34 (Reynolds, J Virol 2001). We revised the manuscript to incorporate these facts.

There is no labelling for UL31 or UL34 in Fig 7, so it is not clear to which figure the reviewer is referring.

(iv) Fig. 10, can the authors purify and do the same experiment using perinuclear virions? Immunoblot of UL31, UL34, and US3 should be shown in DDX3X depletion.

This is the ultimate experiment we would like to do. We have been trying to do that for the last two decades. So far, only one lab has managed that, but with a significant level of contamination (Padula, J Virol 2009).

The immunoblots for US3 is already shown in the figure. We did not probe UL31 and UL34 since they are absent from mature virions (Reynolds, J Virol 2002; Loret, J Virol 2008). This was indicated in the manuscript.

(v) Throughout the manuscript, the authors analyzed the role of DDX3X using a knockdown experiment. As mentioned in the introduction, DDX3X is the RNA helicase, and it should be better to show the importance of this enzymatic activity and the role of RNA during nuclear egress. A similar system was also observed during a large RNP transport in uninfected *Drosophila* cells. Thus, it would be much more interesting for broader readers if the authors showed the role of RNA during nuclear egress in viral capsid.

Indeed, but this is a completely different topic that we hope to address.

Minor comments.

(i) In Line 34, the word “highjack” is unsuitable. The data in this manuscript shows the importance of DDX3X for viral replication but does not show that viruses utilize the function of DDX3X.

We changed "highjacks" for "redirects".

(ii) Lines 168-169, please indicate UL47 also precipitated as it is essential for nuclear egress of HSV-1 (Liu et al., JVI, 2014). Also, please indicate that UL34 was not detected as UL31 and UL34 make a stable complex in infected cells.

Very good points. We added these important issues in the discussion (see major ii above).

Reviewer #3 (Remarks to the Author):

Summary

Nuclear egress is an unusual kind of nuclear export that herpesviruses use to transport their capsids from the nucleus into the cytoplasm during infection. Although the main viral proteins that mediate this process have been identified, less is known about the host participants. In this manuscript, Khadivjam et al report that the cellular DEAD-box RNA helicase DDX3X appears to contribute to the nuclear egress of the prototypical herpes simplex virus 1 (HSV-1). This is an unexpected finding because DEAD-box RNA helicases are ATP-dependent enzymes that unwind misfolded RNA to help it refold. But how such activity is involved in the nuclear egress process is not obvious. Nonetheless, the authors present evidence that links DDX3X to HSV-1 nuclear egress. They show that during infection, DDX3X relocates from the cytoplasm to the perinuclear region and that in the absence of DDX3X, the nuclear egress of capsids is reduced. They also show that DDX3X interacts with the viral protein UL31, which is a component of the viral nuclear egress complex (NEC) that mediates the nuclear budding stage of the egress. This interaction, be it direct or indirect, could explain why relocation of DDX3X to the perinuclear region requires UL31. Less clear is the precise role of the DDX3X in nuclear egress. Authors propose that DDX3X facilitates de-envelopment of the perinuclear enveloped virions (PEVs, which are formed as the result of NEC-mediated nuclear budding of the capsids) by promoting the incorporation of the HSV-1 kinase US3 into the PEVs. But this contention is not sufficiently supported by the presented data. The authors also observe large foci on the nuclear periphery that are enriched in the major capsid protein and do not form in the absence of DDX3X, but the nature of these foci remains uncertain. As the result, while the discovery of the potential involvement of a host RNA helicase in HSV-1 nuclear egress is interesting, the proposed mechanism is confusing and is not sufficiently justified. The paper is mostly well written, and many experiments, especially, the confocal images showing co-localisation, are convincing and are done with proper controls. But there are some concerns regarding the experiments. Some important controls are lacking. For example, the authors do not confirm that the DDX3X siRNA treatment, indeed, reduces protein levels. Additionally, several Western Blot images look odd, potentially, due to image manipulation.

Thanks for the positive comments on the quality of the text and well-controlled and

convincing imaging. Thanks also for the detailed review, which clearly improves our manuscript.

Major criticisms

1. Line 78-82: The authors contradict themselves here. If NEC phosphorylation were a negative regulator of nuclear budding, then in the absence of phosphorylation by the US3 kinase, one would expect to see increased rather than decreased budding. Indeed, the accumulation of PEVs, which are the products of budding, is consistent with increased budding. The accumulation of PEVs in this and other viral mutants or host protein knockdowns could be simply due to their overproduction rather than an additional defect in de-envelopment. This is an important point that the authors should consider in interpreting their data and proposing models.

We are merely reporting the literature but agree with the reviewer that some parts are not clear. The latest model, based on recent findings and supported by several labs, suggests that US3 is a negative regulator of capsid budding (e.g. Drs Graham, Banfield, Heldwein). As always, there may be alternative interpretations, but so far the evidence appears reasonable. The reviewer is, of course, right that increased PEV are expected in that scenario. At issue is the second reported role of US3 during virion fusion with the outer nuclear membrane. There must be a controlled balance between these two events but no one knows how. I would keenly read any paper that explains that.

2. Line 132-133, 232-233, 312, and elsewhere: Authors need to confirm the DDX3X knockdown experimentally.

This is, of course, an important point. We have been using the same siDDX3X reagents since 2009 and already reported their efficacy in both knocking down DDX3X in the cell (70% protein inhibition; Khadivjam, J Virol 2017) but also its incorporation in virions (>90%; Stegen, Plos One 2013). We clarified this point in the manuscript.

3. Line 203: The conclusion that the UL31 deletion mutant failed to recruit DDX3X to the nuclear envelope is not justified. Fig. S5B shows a 3-fold reduction in colocalisation. Moreover, currently, it is challenging to assess colocalisation (or lack thereof) in Fig S5. Colocalisation should be quantified as in Fig. 2.

It should be noted that the threefold reduction is statistically significant ($p < 0.01$), but we agree that the word "fail" is too strong and revised it to "recruited significantly less".

Quantification of nuclear ring localisation can be challenging, particularly in the presence of a cytoplasmic staining. The key is to zoom on the images, which we do. High-resolution STED microscopy is a demanding and limiting technique that required special reagents, is difficult and long to perform. Furthermore, it can only be done with two distinct antibodies in our facility due to the physical setup of that microscope. Note also that not all antibodies

work by STED. Finally, it is not essential to quantify co-localisation and would be an overkill. We instead relied on confocal microscopy, a common standard in the literature, to delineate co-localisation.

4. Lines 252-254: The conclusion that the large VP5-containing foci are C-capsid-dependent is not justified. Fig. 7 clearly shows large foci in the Δ 1-50 UL25 mutant even though the number of cells containing such foci is reduced.

The UL25 mutant is not a full deletion mutant, but one that lacks the first 50 amino acids (Cockrell, J Virol 2009 & 2011). It still expresses normal levels of VP5, produces A- and B- nuclear capsids and leaks limited numbers of infectious particles (10^3 vs 10^7 pfu/ml). It is therefore a pretty good control. Importantly, we see a highly significant reduction in the level of VP5 foci ($p < 0.001$) so there is a link between the foci and C-capsids. It may well be that the residual foci represent the limited A- and B- capsids that do escape the nuclei (Rémillard-Labrosse, J Virol 2006). That possibility is now mentioned in the results section.

5. Line 259-260 and Fig. 8A. The nuclear localisation of UL34 in Δ VP5 and Δ 1-50 UL25 mutants is disconcerting. Why is UL34 inside the nucleus? Is it lacking its transmembrane anchor? Even in transfected cells where no other viral proteins are present, UL34 localises to the nuclear envelope in the presence of UL31.

This was disturbing to us as well. After asking my student to repeat these experiments, we had to conclude it is real. Note that we also found UL34 on HSV-1 nuclear capsids by proteomics (i.e. using MS rather than antibodies), thereby orthogonally confirming our data. For now, we can only speculate what this means, but cannot explain this finding for the moment. For example, could UL34 be "protected" from membrane incorporation as isoprenylated Rab proteins are (as mentioned in El Bilali, J Virol 2021) or perhaps a cleaved soluble product of full length protein?

6. Authors report that DDX3X colocalises with VP5 in large foci at the nuclear periphery that are seen in 20% of the infected cells (Fig. 5). These foci disappear when DDX3X is knocked down (Fig. 6). What are these large foci? They are unlikely to be nuclear envelope herniations containing PEVs because these structures increase in the absence of DDX3X (Fig. 3). They can form even in the absence of C-capsids albeit in a smaller number of cells (Fig. 7). In a contradictory manner, the authors then propose that the nuclear envelope herniations containing PEVs (Fig. 9B) correspond to the large foci observed by confocal microscopy. What are these foci? Are they the herniations or not? At any rate, it is important to note that the herniations are very rare and quite small in cells infected with the WT HSV-1.

Very good analysis. Herniations are indeed rare and usually seen in situations where critical components are not functional (e.g. US3, mutated UL31). To be honest, we

expected to find by EM capsid aggregates closely apposed to the inner nuclear membrane. However, our prediction proved wrong and we instead only found perinuclear virions in clear inward herniations. Then we reasoned that DDX3X may act like US3, i.e. during both nuclear budding, then fusion. The fact that DDX3X physically and functionally interacts with the NEC but also recruit US3 to the viral particles makes sense in that scenario. As you likely know, knocking down US3 primarily produces an accumulation of PEVs in perinuclear herniations, as does DDX3X depletion. But we know that US3 functionally interacts with the NEC prior to that step within the nucleus. Our data fairly strongly suggests this is also the case for DDX3X. It is not yet clear how all the pieces of the puzzle fit together. As pointed out by reviewer #2, UL47 is another piece of that puzzle as are UL21 and the cellular phosphatase 1 (Graham, Banfield). I look forward to some very interesting science in the herpes nuclear egress field. I hope you will agree with us that the present manuscript is an interesting addition to that complex picture, even if it does not explain everything.

7. Line 296-297 and 323-325: The conclusion that DDX3X promotes the US3 incorporation into the PEVs or that this facilitates the de-envelopment is not sufficiently supported by the presented data. Co-immunoprecipitation of UL31 and US3 from nuclear membranes (Fig. 11) only tells us that they may interact there, be it directly or indirectly. It does not confirm that DDX3X facilitates the incorporation of US3 into PEVs. Likewise, de-envelopment was not assayed here. PEV accumulation can be due to their overproduction.

To our knowledge, there is no way to specifically assay the de-envelopment step other than by EM, which we did. The incorporated of US3 in the viral particles at the level of the PEVs has been documented long ago. What we now know is that late in infection, i.e. during capsid assembly and egress, DDX3X is primarily nuclear. We already knew that DDX3X is recruited to mature virions (Loret, J Virol 2008). We here show that DDX3X and US3 physically interact (directly or indirectly) at the nuclear membrane, that DDX3X also interacts with the NEC and that DDX3X is required for the optimal incorporation of US3 in mature virions. The simplest interpretation is that this all occurs at the nuclear membranes. We, of course, remain open to any other alternatives that could explain the data. To reflect the concerns of the reviewer, we rephrased the above lines.

8. Western Blot panels in Fig. 4A, 10A, 11A, and 11B look odd. Was this image manipulated in any way? If so, this needs to be explained.

This is a serious criticism that should be substantiated. Why do you think the figures were manipulated? The only modification done to the figures was to adjust the background, which was done for all lanes across each blots in agreement with most journals.

Minor criticisms

9. Line 45-52: What is known about the DDX3X mechanism in the host and in viruses listed here? Can its RNA helicase activity account for its involvement in viral replication? If not, Does DDX3X have any activities other than RNA unwinding? It is important to describe these here because non-helicase activities if any, could provide clues into its potential role in HSV nuclear egress.

The manuscript already cites 25 different papers about the implication of DDX3X for different viruses. The bulk of those papers either imply its helicase activity with respect to RNA metabolism or DDX3X's ability to induce an IFN response. We previously showed that the latter case is not important for HSV-1 since mostly neutralized by the virus (Khadivjam, J Virol 2017). In the remaining cases, there are mostly observations that do not detail mechanisms or the implication of particular DDX3X domains. Our manuscript is really a first in that sense since describing novel functions, interactions and even mapping which part of DDX3X is required to interact with the NEC (NB: it is not the helicase domain).

10. Line 36 and throughout: Eliminate “physical” in reference to protein interactions. Protein interactions can be direct or indirect. The term “physical” is nebulous.

As the reviewer know, "physical" implies binding of molecules to one another. Co-IP is the golden standard to demonstrate this. This would remain true even if an intermediate protein links DDX3X to the NEC or US3. Moreover, we find the term "physical" appropriate to contrast it with a "functional" role, which does not automatically require physical interactions. Nearly all interaction web tools include indirect binding data.

11. Line 54-55: How does DDX3X influence HSV gene expression? Please, clarify.

We previously showed that it impacts the transcription and translation of several viral genes (Khadivjam, J Virol 2017). This was clarified in the text.

12. Line 61-63: All 4 types of capsids are referred to as viral intermediates. Does this refer to an intermediate in the virion assembly? Please, clarify.

In a way, yes. HSV-1 replication is quite complex. It duplicates its genome and assembles novel capsids in the nucleus. This starts with procapsids, which are filled with scaffold proteins, are thermos-unstable, free of nucleic acid and are round. Upon cleavage, the scaffold is partially released from the capsids to make room for the viral genome while the capsid acquires an icosahedral shape. These concomitant processes are however not efficient and abortive viral intermediates are produced (A-no scaffold and B-with scaffold) in addition to the normal genome containing C capsids. This was clarified in the text.

13. Line 75: What is meant here by “the smooth distribution of the NEC components

around the nucleus”? If this is in reference to the uneven distribution of the immunofluorescent signal of the NEC components in the absence of US3, then this reflects the accumulation of the NEC-coated PEVs in perinuclear herniations rather than the relocalisation of UL31 and UL34 along the nuclear envelope. Please, clarify.

It does refer to the distribution of NEC in normal or US3 deficient conditions and uses the nomenclature used in the herpes literature. This is complicated by the fact that US3 phosphorylates both UL31 and UL34 and that UL31 mutations that mimic phosphorylation also impact NEC distribution. No one has shown if the NEC is present on PEVs in the absence of US3.

14. Line 82-83: Several proteins, e.g., SLC35E1, are missing from the list of host proteins that have been implicated in herpesvirus nuclear egress. ESCRT is not a protein but a class of protein complexes, ESCRT-0, -I, -II, and -III. Which component is referred to here?

SLC35E1 has been added. Regarding the ESCRT, we were referring to the Ari Nature Comm 2018 paper. We now specified the components involved (Alix, CHMP4B).

15. Line 83-84. Unnatural sentence. Please, rephrase.

Done.

16. Line 89: replace “let” with “led”.

Done.

17. Line 94, 128, and elsewhere: replace “repositioning” with “relocalisation”.

Done.

18. Line 95 and elsewhere: none of the methods used in this work establish that the DDX3X and UL31 interact directly, so this should be stated.

As stated above, we do show an interaction, which remains true even if indirect. We do not mention direct binding.

19. Line 100: replace virions with PEVs or an equivalent term, to distinguish these intermediates from mature, infectious virions.

Done.

20. Line 101-102: Is DDX3X present in PEVs?

We do not know. We tried immuno-EM but the antibodies did not work.

21. Line 110: Replace “nucleus” with “nuclear envelope”.

Done.

22. Line 127, 140-141, 296-297, 301-302, and elsewhere: The conclusion that DDX3X modulates PEV de-envelopment is not supported by the data. De-envelopment was not assayed here. PEV accumulation can be due to their overproduction (see comment 1). The presented data only show that in the absence of DDX3X, just as in the absence of US3, PEVs accumulate.

Done.

23. Line 134: Surely, extracellular virions are not viral assembly intermediates.

Fixed!

24. Line 150-164: Here, it would be helpful to briefly explain that this was a pulldown experiment.

Done.

25. Line 172-174. Was this a pull down?

Yes, as already detailed in the manuscript.

26. Line 183: What is meant by “partial” interaction here? How was this assessed? The colocalisation shown in Fig. S3 should be quantified as in Fig. 2.

It simply means partial co-localisation, which was clarified in the text. Quantification is not really required to convince the reader that there is partial-localisation as the figure speaks for itself.

27. Lines 185-197: The binding experiment setup needs to be explained upfront.

This is already detailed in the figure legend but we added extra details in the text.

28. Line 209: Explain what is meant by the “smooth positioning” here.

See above.

29. Line 215: This section is very long and hard to follow. Consider breaking it up into several sections to increase readability.

Great suggestion. We divided up that section.

30. Line 222-235: Authors report that DDX3X colocalises with VP5 in large foci at the nuclear periphery that are seen in 20% of the infected cells, but in Fig 5B, not all such foci appear to have a clear DDX3X signal. Colocalisation should be quantified.

Fig 5B shows that nearly all peripheric VP5 large foci are indeed positive for DDX3X and vice versa. The remaining VP5 foci are within the nucleus, as best seen in the merge images. We nonetheless quantified 6 individual cells from each of the three independent experiment and found that 100% of the VP5 large foci were DDX3X positive. This was added to the results section.

31. Line 235-242: What kind of capsid transport or aggregate transport is measured here (Fig. 6C)? Please, explain the rationale and the results better.

We meant whether DDX3X alters the proximity of the large VP5 foci with the nuclear membrane. At issue was whether DDX3X mediates not only the formation but also the distribution of the foci in the nucleus. We clarified this in the manuscript.

32. Lines 305-308: The rationale for assaying extracellular virions for tegument proteins as a readout for nuclear egress is puzzling. Tegument proteins are present in the cytoplasm and can be recruited there. Even if the amount of US3 in extracellular virions is somewhat lower in the absence of DDX3X (Fig. 10), this does not tell us anything about its incorporation into the PEVs.

Most teguments are indeed recruited to the viral particles in the cytoplasm. The optimal assay would have been to test the presence of various teguments in the PEVs. As you may be aware, no one but one lab succeeded to purify that viral intermediate, but purity was clearly an issue (Padula, J Virol 2009). On the other hand, immuno-EM is very tedious especially to test several proteins, let alone to find antibodies that work. The next possible option was therefore to probe if the lack of DDX3X had an impact on the end product of viral egress, i.e. mature extracellular virions. We clarified these points in the manuscript. Fortunately, we know from the literature that US3 is incorporated in the viral particles at the level on the nucleus. This was merely good luck that we found that DDX3X impacts US3 virion incorporation.

33. There are too many figures. Consider combining Figs 1 and 2, 10 and 11, S1 and S2, and moving 6A, 6C, and 7A into SI.

We kept the essential elements in the main manuscript. We already have 8 figures as supplementary data. The two other reviewers did not have an issue with this.

34. Fig. 1: the size of the nucleus in the 9h time point appears larger. Is the scale the same for all timepoints?

Oups our mistake! The scale bar is distinct in the different panels. We added the scale bars (all at 5 μm).

35. Fig. 3: In panels AB, consider showing boxes in the zoomed-out panels to clarify where magnified images come from. In panel C, what do N1 and N2 refer to?

Boxes were added to indicate the zoom-out sections. N # refers to the two independent experiments. This was clarified in the legend.

36. Fig 4: Was a reverse co-immunoprecipitation using HA-tagged pUL31 done? Additionally, the DDX3X bands in 4C do not match those in Fig S4.

Reciprocal IP were done in panels A and B. There was a shift in the MW markers. This has been corrected.

37. Fig 5: does the DDX3X colocalise with gB within the large foci?

Yes. There is a partial co-localisation, as seen by the yellow labeling.

38. Fig 9: Two different WT stains KOS and 17+ were used in different experiments. The authors need to confirm that both strains behave similarly with regard to the phenotypes described here.

They do. In fact, we used three different strains (KOS, 17⁺ and F). For most experiments strain 17⁺ was used as our standard lab strain, except when viral mutants were tested. For instance, NEC mutants are based on strain F, while VP5 and UL25 mutants are KOS based. The difference stems from the fact these mutants were generated in other laboratories as detailed in the Methods section. We clarified the strain of each mutant in the Methods section and fixed strain issues in some legends.

It should be noted that DDX3X nuclear localisation was noted in strains 17⁺ (figs 1, 2 and 3), F (figs S3, S5 and S6) as well as KOS (fig 8). We also observed large VP5 foci in the three strains (strain 17⁺: fig 5, 6; strain F: S7 and 9; strain KOS: fig 7). Meanwhile the MS and associated data showing an interaction between DDX3X and the NEC was done in strain 17⁺. It is therefore pretty clear that the observed phenotypes are not strain specific. This was clarified in the results and discussion.

REVIEWERS' COMMENTS:

Reviewer #1 (Remarks to the Author):

The authors have satisfactorily answered all the queries raised.

Reviewer #2 (Remarks to the Author):

I am satisfied with the modification of the manuscript.

Reviewer #3 (Remarks to the Author):

The authors have addressed my criticisms.